**Temporal characteristics of atmospheric ammonia and nitrogen dioxide over China based on**
**emission data, satellite observations and atmospheric transport modeling since 1980**
Lei Liu [a], Xiuying Zhang [a, *], Wen Xu [b], Xuejun Liu [b], Yi Li [c], Xuehe Lu [a], Yuehan Zhang [d], Wuting
Zhang [a, e]
[a] Jiangsu Provincial Key Laboratory of Geographic Information Science and Technology, International
Institute for Earth System Science, Nanjing University, Nanjing 210023, China
[b] College of Resources and Environmental Sciences, Centre for Resources, Environment and Food
Security, Key Lab of Plant-Soil Interactions of MOE, China Agricultural University, Beijing 100193,
China
[c] Air Quality Division, Arizona Department of Environmental Quality, Phoenix, AZ, 85007, USA
[d] School of Atmospheric Sciences, Nanjing University, Nanjing, China
[e] Jiangsu Center for Collaborative Innovation in Geographical Information Resource Development and
Application, Nanjing 210023, China
* Corresponding authors: Xiuying Zhang (lzhxy77@163.com)
**Abstract**
China is experiencing intense air pollution caused in large part by anthropogenic emissions of reactive
nitrogen (Nr). Atmospheric ammonia ($NH_3$) and nitrogen dioxide ($NO_2$) are the most important
precursors for Nr compounds (including $N_2O_5$, $HNO_3$, HONO and particulate $NO_3^-$ and $NH_4^+$) in the
atmosphere. Understanding the changes of $NH_3$ and $NO_2$ has important implications for the regulation
of anthropogenic Nr emissions, and is a requirement for assessing the consequence of environmental
impacts. We conducted the temporal trend analysis of atmospheric $NH_3$ and $NO_2$ on a national scale
since 1980 based on emission data (during 1980-2010), satellite observations (for $NH_3$ since 2008 and
for $NO_2$ since 2005) and atmospheric chemistry transport modeling (during 2008-2015).
Based on the emission data, during 1980-2010, both significant continuous increasing trend of $NH_3$ and
$NO_x$ were observed from REAS (Regional Emission inventory in Asia, for $NH_3$ 0.17 kg N $ha^{-1}$ $y^{-2}$ and
for $NO_x$ 0.16 kg N $ha^{-1}$ $y^{-2}$) and EDGAR (Emissions Database for Global Atmospheric Research, for
$NH_3$ 0.24 kg N $ha^{-1}$ $y^{-2}$ and for $NO_x$ 0.17 kg N $ha^{-1}$ $y^{-2}$) over China. Based on the satellite data and
atmospheric chemistry transport modeling named as the Model for Ozone and Related chemical
Tracers, version 4 (MOZART-4), the $NO_2$ columns over China increased significantly from 2005 to
2011 and then decreased significantly from 2011 to 2015; the satellite-retrieved $NH_3$ columns from
2008 to 2014 increased at a rate of 2.37% $y^{-1}$. The decrease in $NO_2$ columns since 2011 may result from
more stringent strategies taken to control $NO_x$ emissions during the 12th Five-Year-Plan, while no
control policy focused on $NH_3$ emissions. Our findings provided an overall insight on the temporal
trends of both $NO_2$ and $NH_3$ since 1980 based on emission data, satellite observations and atmospheric
transport modeling. These findings can provide a scientific background for policy-makers that are
attempting to control atmospheric pollution in China. Moreover, the multiple datasets used in this study
have implications for estimating long-term Nr deposition datasets to assess its impact on soil, forest,
water and greenhouse balance.
**Keywords**: trends, seasonal cycle, ammonia
**1. Introduction**
Reactive nitrogen (Nr) emissions have increased significantly in China due to anthropogenic activities
such as increased combustion of fossil fuels, over-fertilization and high stocking rates of farm animals
(Canfield et al., 2010;Galloway et al., 2008;Liu et al., 2013). Elevated Nr in the environment has led to
a series of effects on climate change and ecosystems, e.g. biodiversity loss, stratospheric ozone
depletion, air pollution, freshwater eutrophication, the potential alteration of global temperature,
drinking water contamination, dead zones in coastal ecosystems and grassland seed bank depletion
(Basto et al., 2015;Lan et al., 2015;Shi et al., 2015). Atmospheric reactive N emissions are dominated
by nitrogen oxides ($NO_x = NO + NO_2$) and ammonia ($NH_3$) (Li et al., 2016a;Galloway et al., 2004).
Atmospheric $NO_2$ and $NH_3$ are the most important precursors for Nr compounds including $N_2O_5$, $HNO_3$,
HONO and particulate $NO_3^-$ and $NH_4^+$ in the atmosphere (Xu et al., 2015;Pan et al., 2012). Therefore,
an understanding of both the spatial and temporal patterns of $NO_2$ and $NH_3$ is essential for evaluating
N-enriched environmental effects, and can provide the scientific background for N pollution mitigation.
To investigate the spatial and temporal variations of atmospheric $NO_2$ and $NH_3$, ground measurements
are acknowledged to be an effective way in monitoring the accurate concentrations of $NO_2$ and $NH_3$
(Xu et al., 2015;Pan et al., 2012;Meng et al., 2010). Ground measurements of $NO_2$ concentrations in
China, including about 500 stations in 74 cities, have been monitored and reported to the public since
January 2013 (Xie et al., 2015). By the end of 2013, this network was extended with hourly $NO_2$
concentrations from more than 850 stations in 161 cities. However, there are fewer $NH_3$ measurements
across China than $NO_2$ measurements. The China Agricultural University has organized a Nationwide
Nitrogen Deposition Monitoring Network (NNDMN) since 2010, consisting of 43 monitoring sites
covering urban, rural (cropland) and background (coastal, forest and grassland) areas across China (Xu
et al., 2015;Liu et al., 2011). Xu et al. (2015) reported the ground $NH_3$ concentrations throughout China
for the first time, providing great potential to understand the ground $NH_3$ concentrations on a national
scale. Other networks include (1) the Chinese Ecosystem Research Network (CERN) which was
established in 1988, including 40 field stations (Fu et al., 2010). However, to our knowledge, there are
no detailed reports about ground $NH_3$ concentrations from CERN on a national scale. (2) Four Chinese
cities (Xiamen, Xi-An, Chongqing and Zhuhai) have joined the Acid Deposition Monitoring Network
in East Asia (EANET) since 1999. However, only one site (Hongwen, Xiamen) in EANET measured
the ground $NH_3$ concentrations and that data is not continuous. Finally, ground $NH_3$ concentrations at
ten sites in Northern China from 2007 to 2010 have been reported by Pan et al. (2013). All of the above
ground measurements provide the potential to understand $NH_3$ and $NO_2$ concentrations on a regional
scale. However, there is limited information on the spatial and temporal variations of $NH_3$ and $NO_2$ in
the atmosphere across China. This is due to the limited observation sites and monitoring period, as well
as given the uneven distribution of the monitoring sites. Importantly, atmospheric $NH_3$ and $NO_2$
monitoring based on ground-based local sites may have limited spatial representativeness of the
regional scale as both $NH_3$ and $NO_2$ are highly variable in time and space (Clarisse et al., 2009;Wichink
Kruit et al., 2012;Boersma et al., 2007).
In order to complement ground-based measurements, satellite observation of $NH_3$ and $NO_2$ is a
welcome addition for analyzing the recent trends of $NH_3$ and $NO_2$ in the atmosphere. Satellite remote
sensing offers an opportunity to monitor atmospheric $NH_3$ and $NO_2$ with high temporal and spatial
resolutions (Warner et al., 2017;Li et al., 2016b). $NO_2$ was measured by multiple space-based
instruments including the Global Ozone Monitoring Experiment (GOME), SCanning Imaging
Absorption SpectroMeter for Atmospheric CHartographY (SCIAMACHY), Ozone Monitoring
Instrument (OMI) and Global Ozone Monitoring Experiment-2 (GOME-2). The OMI $NO_2$ provides the
best horizontal resolution ($13 \times 24$ $km^2$) among instruments in its class and near-global daily coverage
(Levelt et al., 2007). OMI observations have been widely applied in environmental-related studies and
for the support of emission control policy (Russell et al., 2012;Zhao and Wang, 2009;Castellanos et al.,
2015;Lamsal et al., 2015;Liu et al., 2016a;Foy et al., 2016). First measurements of $NH_3$ from space
were reported over Beijing and San Diego areas with the Tropospheric Emission Spectrometer (TES)
(Beer et al., 2008) and in fire plumes in Greece with the Infrared Atmospheric Sounding Interferometer
(IASI) (Coheur et al., 2009). The first global map of $NH_3$ was created from IASI measurements by
correlating the observed brightness temperature differences to $NH_3$ columns using the averaged
datasets in 2008 (Clarisse et al., 2009). Shortly after that, many studies focused on developing
techniques to gain more reliable $NH_3$ columns (Whitburn et al., 2016a;Van Damme et al., 2014b),
validating the retrieved $NH_3$ columns using the ground measurements (Van Damme et al.,
2014a;Dammers et al., 2016) and comparing the data with the results of the atmospheric chemistry
transport models (Van Damme et al., 2014c;Whitburn et al., 2016a), and the estimated $NH_3$ columns
obtained from Fourier transform infrared spectroscopy (FTIR) (Dammers et al., 2016). The retrieval
algorithm of obtaining IASI $NH_3$ columns was based on the method described in Whitburn et al. (2016).
Two main steps were performed to derive the $NH_3$ columns from the satellite measurements. First,
derive the spectral hyperspectral range index (HRI) based on each IASI observations (Walker et al.,
2011;Van Damme et al., 2014b). Second, convert HRI to $NH_3$ columns based on a constructed neural
network with input parameters including vertical $NH_3$ profile, satellite viewing angel, surface
temperature and so on (Whitburn et al., 2016a). The progresses made on the satellite techniques
provide possibility for understanding both the spatial and temporal variations of $NH_3$ and $NO_2$ in the
atmosphere.
In addition to satellite observations, the emission data are also very important for investigating the
temporal trends of $NH_3$ and $NO_2$ such as the IIASA inventory (Cofala et al., 2007), EDGAR (Emission
Database for Global Atmospheric Research, RAINS-Asia (Regional Air Pollution Information and
Simulation) and Asia REAS (Regional Emission inventory in Asia). REAS is considered as the first
inventory by integrating historical, current and future emissions data for Asia based on a consistent
methodology (Ohara et al., 2007), and EDGAR is the global emission data with 0.1 by 0.1 grid, which
has the highest spatial resolution among different datasets mentioned above. Thus, REAS and EDGAR
are used to analyze the historical trends of $NH_3$ and $NO_2$ during 1980-2010 in this study. Based on the
EDGAR emission data, a widely used atmospheric transport model named as the Model for Ozone and
Related chemical Tracers, version 4 (MOZART-4) was also used to model the temporal trend of $NH_3$
and $NO_2$ columns during 2008-2015 in comparison with the temporal trends of $NH_3$ and $NO_2$ columns
measured by satellite instruments.
We aim at getting an overall insight on the temporal trends of both $NO_2$ and $NH_3$ since 1980 based on
the multiple datasets including the emission data, satellite observations and atmospheric transport
modeling. We herein show the Chinese national trend of REAS and EDGAR $NH_3$ and $NO_x$ emission
data during 1980-2010, satellite-retrieved $NH_3$ during 2008-2015 and $NO_2$ columns (2005-2015), and
atmospheric transport chemistry modeling $NH_3$ and $NO_2$ columns (2008-2015). It should be noted here
that the satellite $NH_3$ columns were retrieved from the IASI, and can only be obtained since 2008. It is
beneficial to analyze the temporal variations of both $NH_3$ and $NO_2$, hence providing a scientific basis
for policy makers to reduce N-enriched environmental pollution in China.
**2. Materials and methods**
**2.1. $NH_3$ and $NO_2$ Emissions**
We examined the emission inventory dataset for Asia REAS (Regional Emission inventory in Asia)
with 0.5 °× 0.5 °resolution for the period 1980-2010, and analyzed the temporal trends of $NO_x$ and $NH_3$
over China. REAS v1.1 is believed to be the first inventory of integrating past, present and future
dataset in Asia based on a consistent methodology. The REAS datasets have been validated by several
emissions, and denote agreement with the recent growth status in Chinese emissions (Ohara et al.,
2007). We also collected $NO_x$ and $NH_3$ emission data from EDGAR (Emissions Database for Global
Atmospheric Research) v4.3.1, which was developed by the Netherlands Environmental Assessment
Agency and European Commission Joint Research Centre (Jgj et al., 2002). The EDGAR emissions are
calculated on the basis of a point emissions inventory conducted by the International Energy Agency.
EDGAR also has a long time period 1980-2010 with the highest spatial resolution globally (0.1 °×0.1 °)
(http://edgar.jrc.ec.europa.eu/overview.php?v=431).
**2.2. Satellite observations**
IASI is a passive remote-sensing instrument operating in nadir mode and measures the infrared
radiation emitted by the Earth's surface and the atmosphere (Clarisse et al., 2009). It covers the entire
globe twice a day, crossing the equator at a mean solar local time of 9:30 A.M. and P.M. and has an
elliptical footprint of 12 by 12 km up to 20 by 39 km depending on the satellite-viewing angle. In this
study we use daytime satellite observations as these are more sensitive to $NH_3$ and are associated with a
large positive thermal contrast and a significant amount of $NH_3$ (Van Damme et al., 2014b;Whitburn et
al., 2016a). The availability of measurements is mainly driven by the cloud coverage as only
observations with cloud coverage lower than 25% are processed to be a good compromise between the
number of data kept for the analysis and the bias due to the effect of clouds. As the amount of daily
data is not always sufficient to obtain meaningful distributions (due to cloud cover or the availability of
the temperature profiles from the EUMETSAT operational processing chain) (Van Damme et al.,
2014b), it is more appropriate to consider monthly or yearly averages for this trend analysis. We
consider IASI observations with a relative error below 100% or an absolute error below $5\times10^{15}$ molec.
cm$^{-2}$ for analysis over China. For the error, the filtering depends on the use of the data. Doing this, low
columns typical for background conditions with a large relative error but a small absolute error are also
taken into account. For other applications, such as comparing with ground measurements, we would
recommend to use a threshold of 75% or even 100% relative error. We gained the data upon request
from the Atmospheric Spectroscopy Group at Université Libre De Bruxelles
(http://www.ulb.ac.be/cpm/atmosphere.html). This data can be gridded on 0.1 ° latitude×0.1 ° longitude
(Dammers et al., 2016), 0.25 ° latitude×0.25 ° longitude (Whitburn et al., 2016a) and 0.5 ° latitude×0.5 °
longitude (Whitburn et al., 2016b) or even coarser resolutions depending on the usage of the data. For
IASI NH$_3$, we firstly divided China into 0.5 ° latitude×0.5 ° longitude grid. For each grid cell, we
calculated the monthly arithmetic mean by averaging the daily values with observations points within
the grid cell. Similarly, we calculated the annual arithmetic mean by averaging the daily values with
observations points within the grid cell over the whole year.
The NO$_2$ columns are obtained from the OMI instrument on NASA's EOS Aura satellite globally
everyday. We used the generated products by the project "Derivation of Ozone Monitoring Instrument
tropospheric NO$_2$ in near-real time" (DOMINO) to analyze the temporal trends of NO$_2$ columns over
China. In DOMINO products, only the observations with a cloud radiance fraction below 0.5 were
processed for analysis. The retrieval algorithm is described in detail in the previous work (Boersma et
al., 2007) and recent updates can be found in the DOMINO Product Specification Document
(http://www.temis.nl/docs/OMI_NO2_HE5_1.0.2.pdf). We used tropospheric NO$_2$ retrievals from the
DOMINO algorithm v2.0. The retrieval quality of NO$_2$ products is strongly dependent on different
aspects of air mass factors, such as radiative transfer calculations, terrain heights and surface albedo.
The OMI v2.0 data were mainly improved by more realistic atmospheric profile parameters, and
include more surface albedo and surface pressure reference points than before (Boersma et al.,
2011;Boersma et al., 2016). The DOMINO $NO_2$ datasets are available from
http://www.temis.nl/airpollution/no2.html. We should state in particular that we used directly the
DOMINO v2.0 products of monthly means from 2005 to 2015 over China for the trend analysis. The
DOMINO $NO_2$ columns were gridded at a resolution of 0.125 °latitude×0.125 °longitude grid globally,
which has been widely used for scientific applications (Ma et al., 2013;Ialongo et al., 2016;Castellanos
et al., 2015).
To illustrate measurement availability, we presented here some measurement statistics. A total number
of cloud-free daytime observations as characterized by the operational IASI processor by year were
retrieved in China during 2008-2015 for $NH_3$ (Fig. 1b). We retrieved more observation numbers after
2010 than those during 2008-2009. In 2010, the update of the improved air temperature profiles, cloud
properties products and cloud detection, which are important for calculating the thermal contrast,
increased the quality of retrieval (Van Damme et al., 2014b;Van Damme et al., 2014c). In September
2014, there was another update of the air temperature profiles, cloud properties products and cloud
detection for calculating the thermal contrast. For the updates of the IASI-$NH_3$ data, you can refer to
Van Damme et al. (2014b), Van Damme et al. (2014c) and Whitburn et al. (2016). The monthly
observation numbers are also presented in Fig. 1a, showing that spring (Mar, Apr and May), summer
(Jun, Jul and Aug), autumn (Sep, Oct and Nov) and winter (Dec, Jan and Feb) months represent 29% ,
26%, 23% and 21%, respectively. Compared with large variations of observation numbers for $NH_3$, the
observation numbers for $NO_2$ varied less by year; winter season had the least, while other seasons
varied little.
**2.3. Atmospheric transport chemistry model**
Atmospheric transport chemistry model is also of central importance in modeling the tropospheric $NO_2$
and $NH_3$. We applied a widely used atmospheric global atmospheric transport chemistry model named
as the Model for Ozone and Related chemical Tracers, version 4 (MOZART-4) to simulate the
tropospheric $NO_2$ and $NH_3$ columns during 2008-2015 in accordance with the time period of IASI $NH_3$
measurements.
The MOZART-4 model is driven by the meteorological data from the NASA Goddard Earth Observing
System Model, Version 5 (GEOS-5) at a resolution of $1.9\,°$ latitude $\times\, 2.5\,°$ longitude spatially. The
emission data applied for driving the simulations are based on the updated EDGAR emission
inventories. 12 bulk aerosol compounds, 39 photolysis, 85 gas species as well as 157 gas-phase
reactions were integrated in MOZART-4. The chemical mechanism on N compounds including the $NO_2$,
$NH_3$ and aerosols are detailedly integrated to MOZART-4, which is considered to be suitable for
tropospheric chemical compositions (Emmons et al., 2010;Pfister et al., 2008;Sahu et al., 2013). The
output data used in the current work are temporally varying six hours every day, which were upon
request by Louisa Emmons at National Center for Atmospheric Research (NCAR). The monthly means
of $NO_2$ and $NH_3$ columns were averaged by the daily data, and then used for the trend analysis over
China. For more details about MOZART-4, the reader should refer to previous studies (Emmons et al.,
2010;Brasseur et al., 1998;Beig and Singh, 2007).
**3. Results and discussions**
**3.1. $NH_3$ and $NO_2$ emissions during 1980-2010**
We conducted the temporal analysis of $NH_3$ and $NO_x$ emissions since 1980 based on REAS and
EDGAR. Both significant continuous increasing trends of $NH_3$ and $NO_x$ were observed from REAS
(for $NH_3$ 0.17 kg N $ha^{-1}$ $y^{-2}$ and for $NO_x$ 0.16 kg N $ha^{-1}$ $y^{-2}$) and EDGAR (for $NH_3$ 0.24 kg N $ha^{-1}$ $y^{-2}$
and for $NO_x$ 0.17 kg N $ha^{-1}$ $y^{-2}$) over China (Fig. 2). We found a relatively consistent increase in $NO_x$
emission from EDGAR and REAS over China, i.e. 0.17 kg N $ha^{-1}$ $y^{-2}$ vs 0.16 kg N $ha^{-1}$ $y^{-2}$, but
inconsistency in the magnitude of $NH_3$ emissions from EDGAR and REAS over China, i.e. 0.24 kg N
$ha^{-1}$ $y^{-2}$ vs 0.17 kg N $ha^{-1}$ $y^{-2}$. The increase rate in $NH_3$ emissions over China from EDGAR was much
higher than that from REAS, indicating the magnitude of increase trend in $NH_3$ over China remains a
debate, although their thread values of 0.24 kg N $ha^{-1}$ $y^{-2}$ (EDGAR) vs 0.17 kg N $ha^{-1}$ $y^{-2}$ (REAS) both
reflected a continuous increasing trend (in this regard they are consistent). It implies that, at least, the
$NH_3$ emissions are indeed increasing during 1980-2010. We also conducted a simple correlation
analysis of the $NH_3$ (Fig. 2a) and $NO_x$ (Fig. 2b) from REAS and EDGAR, showing agreement in the
magnitude (slope=1.06) and temporal trend ($R^2$=0.96) for $NO_x$, but some inconsistency in the increase
rate (slope=1.33) for $NH_3$.
The discrepancy in the magnitude of $NH_3$ increase rate from REAS and EDGAR (0.24 kg N $ha^{-1}$ $y^{-2}$ vs
0.17 kg N $ha^{-1}$ $y^{-2}$) in China since 1980 may be caused by the different emission factors considered for
estimating $NH_3$ emissions. The EDGAR v4.3.1 $NH_3$ emissions were calculated based on a variety of
sectors including agriculture, shipping, waste solid and wastewater, energy for buildings, process
emissions during production and application, power industry, oil refineries, transformation industry,
combustion for manufacturing, road transportation, railways, pipelines and off-road transport, while the
REAS v1.1 $NH_3$ emissions focused mainly on the agriculture source (i.e., manure management of
livestock and fertilizer application) (Crippa et al., 2015;Ohara et al., 2007). Moreover, the fundamental
methodology on estimating the REAS v1.1 $NH_3$ emissions did not consider the seasonal agricultural
variations compared with that of EDGAR v4.3.1 $NH_3$ emissions (Kurokawa et al., 2013), and the
removal efficiency (as a key element to estimate $NH_3$ emissions) was also reported to be much higher
in REAS v1.1 than in EDGAR v4.3.1 (Kurokawa et al., 2013).
A previous study (Liu et al., 2013) summarized published data on the national anthropogenic $NH_3$ and
$NO_x$ emissions with multi-periods in China (Wang et al., 2009;Wang et al., 1997;Streets et al.,
2003;Klimont et al., 2001;Sun and Wang, 1997;Olivier et al., 1998;FRCGC, 2007), and also analyzed
the temporal pattern of $NH_3$ emissions. Their results showed that the $NH_3$ emissions had increased at an
annual average rate of 0.32 Tg N $y^{-2}$ (about 0.33 kg N $ha^{-1}$ $y^{-2}$). The increase rate of $NH_3$ emissions
(0.33 kg N $ha^{-1}$ $y^{-2}$) by Liu et al. (2013) was double that in REAS (0.17 kg N $ha^{-1}$ $y^{-2}$), implying that the
$NH_3$ increase rate in China is still an open question, and should be further studied.
**3.2. Satellite $NH_3$ and $NO_2$ over China in the recent decade**
**3.2.1. Temporal trends**
We referred to the method of a previous study (Russell et al., 2012) to conduct the temporal trend
analysis by calculating the average values during cold months (October-March) and warm months
(April-September) respectively. We herein concentrated more on the temporal analysis of satellite
observations during warm months because of the relatively lower uncertainty in comparison with that
during cold months. Fig. 3 shows the temporal trend of $NO_2$ columns during warm and cold months
between 2005 and 2015 as well as monthly average values. From satellite observations, the $NO_2$
columns over China increased with a slope of $0.063 \times 10^{15}$ molec. $cm^{-2}$ $y^{-1}$ (4.07% $y^{-1}$) in warm months
from 2005 to 2011 and then decreased with a slope of -0.072 molec. $cm^{-2}$ in warm months (-3.62% $y^{-1}$)
from 2011 to 2015 (Fig. 3 ). The decreasing trends were consistent with $NO_x$ emissions since 2011 over
China (decreasing from $24.04 \times 10^6$ ton in 2011 to $20.78 \times 10^6$ ton in 2014, China Statistical Yearbook,
http://www.stats.gov.cn/). During the Chinese 11th Five-Year-Plan (FYP) period (2006-2010), Chinese
government undertook a series of strategies to increase energy efficiency and to reduce $NO_x$ emissions,
but $NO_x$ emissions were not successfully restrained, which created a big challenge for improving air
quality over the country (Xia et al., 2016). During the 12th FYP period (2011-2015), more stringent
strategies were implemented to control $NO_x$ emissions, including the application of selective
catalytic/non-catalytic reduction (SCR/SNCR) systems in the power sector, staged implementation of
tighter vehicle emission standards and a series of standards with aggressive emission limits for power,
cement, and the iron and steel industries. These strategies are believed to have helped achieve national
targets of $NO_x$ emission abatement (Xia et al., 2016).
However, the satellite-retrieved $NH_3$ columns increased with a slope of $0.118 \times 10^{15}$ molec. $cm^{-2}$ $y^{-1}$
(2.37% $y^{-1}$) in warm months from 2008 to 2014 (Fig. 3), but increase largely in 2015 (this will be
discussed in Sect. 3.3 in comparison with MOZART-4 simulations in detail). The percent increase rate
for $NH_3$ by year (2.37% $y^{-1}$) from 2008 to 2014 is lower than that for $NO_2$ (4.07% $y^{-1}$) from 2005 to
2011, although the absolute $NH_3$ increase rate of $0.118 \times 10^{15}$ molec. $cm^{-2}$ $y^{-1}$ from 2008 to 2014 was
higher than absolute $NO_2$ increase rate of $0.063 \times 10^{15}$ molec. $cm^{-2}$ $y^{-1}$ from 2005 to 2011. An increase in
$NH_3$ columns from IASI may be due to decreased $NH_3$ removal leading to a larger fraction maintaining
in gaseous state for a long time rather than changing to the condensed phase. Specifically, $NH_3$ is
considered as an important alkaline gas that is abundant in the atmosphere, and is able to neutralize
acidic components including $HNO_3$ and $H_2SO_4$ through the oxidation of $NO_x$ and $SO_2$, respectively (Li
et al., 2014;Liu et al., 2011;Liu et al., 2017c;Xu et al., 2015). The decreased $NH_3$ removal to some
degree can be attributed to continuous decreased acidic gases including the $NO_2$ and $SO_2$ over China
under strong control policy in 12-th FYP, which can largely decrease the fraction of the chemical
conversion to $(NH_4)_2SO_4$ and $NH_4NO_3$ in the atmosphere. Increasing trend in $NH_3$ columns may be
associated with continuous N fertilizer use for guaranteeing increase of crop productions (Erisman et
al., 2008). Although there was no strong $NH_3$ emission control regulation, N fertilizer efficiency should
be further improved over China. In 2015, the Ministry of Agriculture formally announced a "Zero
Increase Action Plan" for national fertilizer use by 2020, which requires the annual increase in total
fertilizer use will be less than 1% from 2015 to 2019, with no further increment from 2020 (Liu et al.,

290    2015).

If the "Zero Increase Action Plan" for N fertilizer can be effective, future $NH_3$ emissions should be
consistent with the current $NH_3$ emissions. In addition, due to strong emission control of $NO_x$, the $NO_x$
emissions were believed to decrease significantly from 2011 to 2015. We can reasonably make two
major conclusions. First, the atmospheric $NO_2$, as a key indicator of oxidized N compounds ($NO_2$,
$HNO_3$ and $NO_3^-$), decreased since 2011, and will continue to decrease under the current policy. Second,
the atmospheric $NH_3$, as a key indicator of reduced N ($NH_3$ and particulate $NH_4^+$), will slightly increase
or stay at the current level in the future with the "Zero Increase Action Plan". Thus, due to a decreasing
trend of oxidized N ($NO_x$-N), ammonia N ($NH_x$-N) should still dominate Nr deposition (oxidized N
plus reduced N) in China, and is expected to play a more significant role in Nr deposition. Therefore,
monitoring the reduced N on a regional scale is encouraged to assist in enacting effective measures to
protect the environments and public health, with respect to air, soil and water quality.
**3.2.2. Spatial pattern**
High $NH_3$ columns were found in Beijing, Hebei, Henan, Shandong, Hubei and Jiangsu provinces and
in Eastern Sichuan province (Fig. 4a), which were consistent with their high $NH_3$ emissions due to
intensive fertilizer application and livestock (Huang et al., 2012). Guangdong, Guangxi, Hunan and
Jiangxi provinces also showed high $NH_3$ columns, due to high volatilization from paddy fields in these
regions, with rice being the dominant crop and contributing the most emissions. High $NH_3$ columns in
southern China are in agreement with the high percent paddy farmland area (Fig. S1a) and the high
$NH_3$ columns in northern China are in agreement with the high percent dry farmland area (Fig. S1b). In
addition, the $NH_3$ emissions from vehicles in urban areas could also contribute to the observed high
$NH_3$ columns. For example, in Beijing, the contribution of vehicles equipped with catalytic converters,
particularly since the introduction of three-way-catalysts, to non-agricultural $NH_3$ emissions has
recently been considered and might be the most important factor influencing $NH_3$ concentrations in
urban cities (Meng et al., 2011;Xu et al., 2017). In addition, Xinjiang province also emits remarkable
$NH_3$ emissions related to sheep manure management (Huang et al., 2012;Kang et al., 2016;Zhou et al.,
2015;Liu et al., 2017a). The lower $NH_3$ columns are located mostly in the Tibet Plateau area, where
there is a minimal amount of arable land and low use of synthetic nitrogenous fertilizers.
$NO_2$ columns (Fig. 4b) show significantly higher values over vast areas covering North China, East
China, and the Sichuan Basin. The $NO_2$ columns also show high values over the Pearl River Delta, the
southern part of Northeast China, and some areas in Northwest China. High $NO_2$ columns are mostly
distributed in populated areas (Fig. S2), where there is a mix of various anthropogenic $NO_x$ sources,
such as vehicles and industrial complexes (Wang et al., 2012;Xu et al., 2015;Meng et al., 2010). It
should be noted that an enhanced emission intensity from transportation is confirmed since 2005, even
with staged implementation of tightened emission standards for on-road vehicles (Wang et al., 2012).
For example, $NO_x$ emissions from transportation grew to 30% for the whole country in 2014, and the
values reached 44%, 55%, and 33% for Beijing, Shanghai, and Guangdong, respectively (Xia et al.,
2016). Therefore, transportation is believed to play an increasingly important role in regional $NO_2$
pollution, especially when emissions from stationary sources are gradually controlled through increased
penetration of selective catalytic/non-catalytic reduction (SCR/SNCR) systems.
**3.2.3. Limitations of satellite observations**
It is difficult to gain whole coverage over China based on the daily data for both IASI $NH_3$ and OMI
$NO_2$. For daily $NO_2$, the spatial coverage gained by OMI were influenced by cloud radiance fractions,
surface albedo, solar zenith angles, row anomaly and so on (Russell et al., 2011;De Smedt et al., 2015).
"Row anomaly" issue resulting from the OMI instrumental problem had an impact on approximately
half of the rows undergoing unpredictable patterns in cross-track directions relying on latitudes and
seasons and prevented obtaining convincing daily product with continuous coverage (Boersma et al.,
2011;Boersma et al., 2016). For $NH_3$, the satellite instruments were strongly dependent on the
meteorological conditions such as cloud fractions or the availability of the temperature profiles (Van
Damme et al., 2014b;Boersma et al., 2011), and we cannot retrieve the whole coverage based on daily
data over China. It will be beneficial to analyze a very local region with enough numbers of
observations, but not appropriate to analyze such large coverage over China.
Facing this big challenge, we used the monthly data for the trend analysis over China. The uncertainty
of DOMINO v2.0 $NO_2$ columns has been well documented in Boersma et al. (2011), and the relative
error is reported lower than 20-30% in East Asian by an improved altitude-dependent air mass factor
look-up table, a more realistic atmospheric profile, an increased number of reference vertical layers and
advanced surface albedos (Boersma et al., 2011). The reader is strongly suggested to refer to Boersma
et al. (2011) for more details on the uncertainty analysis.
The potential uncertainty of IASI $NH_3$ columns resulted from IASI observation instruments and
retrieval algorithms. In this paper, the $NH_3$ datasets were generated based on the recent-updated robust
and flexible $NH_3$ retrieval algorithms, which were designed to overcome some shortcomings of the
current algorithms (Whitburn et al., 2016a). The current algorithms were designed firstly to calculate
the hyperspectral range index (HRI), a measure for the $NH_3$ signature strength in the spectrum, and
then converted to IASI $NH_3$ columns by using the thermal contrast (TC) and lookup tables (LUT) of
(HRI, TC) pair corresponding to $NH_3$ columns. The retrieval of HRIs is strongly dependent on the
amount of $NH_3$ and the thermal state of the atmosphere (Whitburn et al., 2016a). The quality of the
IASI $NH_3$ product has been validated by atmospheric chemistry transport models, ground-based and
airborne measurements, and $NH_3$ total columns obtained with ground-based Fourier transform infrared
spectroscopy (FTIR). A first validation of the IASI $NH_3$ using the LOTOS-EUROS model was
conducted over Europe, indicating the respective consistency of IASI measurements and model
simulations (Van Damme et al., 2014c). A first evaluation of IASI $NH_3$ dataset using ground-based
measurements was made worldwide, presenting consistency with the available ground-based
observations and denoting promising results for evaluation by using independent airborne data (Van
Damme et al., 2014a). A first validation of of IASI $NH_3$ dataset using ground-based FTIR derived $NH_3$
total columns was evaluated, demonstrating a mean relative difference of $-32.4\pm(56.3)\%$, a correlation
r of 0.8 with a slope of 0.73 (Dammers et al., 2016).
**3.3. Atmospheric chemistry transport model $NO_2$ and $NH_3$ columns since 2008**
Satellite $NO_2$ and $NH_3$ columns were observed at overpass time as an instantaneous point in a day (at
9:30 A.M. for IASI $NH_3$ and at 1:45 P.M. for OMI $NO_2$ local time). These instantaneous satellite
observations may not be representative for the temporal trend analysis over China. We further retrieved
the monthly variations of $NO_2$ and $NH_3$ columns since 2008 from MOZART varying 6 hours every day
(00, 06, 12, 18 h). We compared the temporal trend analysis of $NO_2$ from MOZART at 12 h with that
gained from satellite at the overpass time (OMI 1:45 P.M. local time) as well as for $NH_3$.
Fig. 5 shows the $NO_2$ columns at 12:00 during warm and cold months between 2008 and 2015 from
MOZART. The percent increase rate for $NO_2$ columns at 12:00 during warm months (April-September)
between 2008 and 2011 was 4.02% $y^{-1}$ (Fig. 5), which was comparable with that (4.23% $y^{-1}$) derived
from OMI (Fig. 3). During 2011-2015, we found a slightly lower decrease rate (-2.93% $y^{-1}$) in $NO_2$
columns during warm months at 12:00 from MOZART (Fig. 5) than that (-3.62% $y^{-1}$) gained from OMI
at 13:45 (Fig. 3). The temporal variations of $NO_2$ columns at 12:00 from MOZART were generally in
accord with those from OMI at 13:45 P.M. local time. Fig. 5 also demonstrates the average $NO_2$
columns (averaged at 00, 06, 12 and 18 h) during warm and cold months between 2008 and 2015. We
found a close increase rate at 12:00 (4.02%) with that averaged at 00, 06, 12 and 18 h (4.23%) before
2011, as well as a similar decrease rate at 12:00 (-2.93%) and the average (-3.07%), implying that the
temporal trend analysis at 12:00 vs. that averaged at 00, 06, 12 and 18 h can be considered mostly
consistent over China from MOZART.
For $NH_3$, we found the percent increase rate at 12:00 during warm months between 2008 and 2015 was
1.30% $y^{-1}$ from MOZART (Fig. 5), which was lower than that (2.37% $y^{-1}$) from IASI during 2008-2014.
The percent increase rate by daily average (at 00, 06, 12 and 18 h) during warm months between 2008
and 2015 was 1.36% $y^{-1}$ from MOZART (Fig. 5). In 2015, we found a relatively large increase in $NH_3$
columns in China during warm months between 2014 and 2015 (50.45%) from IASI, while an increase
from MOZART was about 8.13% between 2014 and 2015. In MOZART-4, the alkaline gaseous $NH_3$
and the acidic gaseous $NO_2$ (the precursor for $HNO_3$) and $SO_2$ are very important precursors for bulk
$NH_4NO_3$ and $(NH_4)_2SO_4$ particles, which form the primary system of gas-particle partitioning
($NH_3$-$NH_4^+$-$NO_x$-$NO_3^-$-$SO_2$-$SO_4^{2-}$). The chemical shifts between particulate $NH_4NO_3$ and gaseous $NH_3$
and $NO_x$ are correlated with the abundance of $NH_3$ and $NO_x$ and meteorological factors. The decreased
abundance of $NO_x$ between 2011 and 2015 may also contribute to an increase in the $NH_3$ abundance in
the gas stage resulting from decreased conversion to particulate $NH_4NO_3$.
Large difference in the $NH_3$ increase rate in 2015 was found between IASI (50.45%) and MOZART
(8.13%). This may be still an open question on this point, here we only show this two possibilities. We
should clarify in particular we do not aim at validating which is right or wrong from IASI and
MOZART (which may be beyond the discussion in this paper), but the $NH_3$ columns in 2015 indeed
increased both from IASI and MOZART. At the current state, we can, at least, draw a conclusion that
the $NH_3$ columns over China indeed increased in 2015 both from IASI and MOZART, but a debate or
inconsistency exists on the increase rate of the $NH_3$ columns in 2015. We should state in particular
again that the following discussion in this paragraph was all hypothetical and should be tested in the
future work. For IASI $NH_3$ columns, the sharp increase in 2015 over China may be an artifact, which
may be due to an update of the input data. Similar jumps in IASI $NH_3$ increase in 2015 can also be
visible in the USA and European (Fig. 6), indicating that it may be necessary for a recalculation of the
earlier input datasets used for calculating the IASI $NH_3$ columns since September, 2014.
**3.4. Implications for estimating long-term Nr deposition datasets and recommendations for**
**future work**
We found both the $NO_x$ and $NH_3$ over China increased continuously from 1980 to 2010 based on
emissions data from REAS and EDGAR. In recent years, based on satellite observations, we found an
increase of 2.37% $y^{-1}$ in $NH_3$ columns during 2008-2014. We also found high-level $NO_2$ columns over
China from 2005-2011 (4.07% $y^{-1}$) but a decrease from 2011 to 2015 (-3.62% $y^{-1}$). Despite the decline,
the $NO_2$ columns during 2011-2015 were still in high level with an average of $1.87 \times 10^{15}$ molec. $cm^{-2}$
$y^{-1}$ compared with that ($1.65 \times 10^{15}$ molec. $cm^{-2}$ $y^{-1}$) during 2005-2010. Notably, these emissions
certainly lead to the deposition of atmospheric Nr in form of dry and wet processes into aquatic
ecosystems and terrestrial, with implications affecting ecosystem and human health, biological
diversity and greenhouse gas balances (Lu et al., 2016). Hence, it is very crucial to estimate Nr
deposition with high spatiotemporal resolutions in order to drive ecological models such as the
Denitrification-Decomposition (DNDC) model and Integrated BIosphere Simulator (IBIS), to assess its
impact on soil, forest, water and greenhouse balance. Here, we call for a long-term dataset of Nr
depositions both regionally and globally to investigate how the N emissions affect the environment.
Challenge still exits in estimating both the dry ($NO_2$, $HNO_3$ particulate $NO_3^-$, $NH_3$ and particulate $NH_4^+$)
and wet ($NH_4^+$ and $NO_3^-$ in precipitation) depositions for a long-term dataset such as since 1980 or
earlier possibly due to the complex scheme of N transformations and transportation or limited available
data both from emissions, satellites and a limited number of ground measurements.
Satellite observations provide a new perspective of estimating Nr depositions regionally, and have been
used to improve the estimation performance. For example, to improve the modeling performance in dry
gaseous $NO_2$ depositions from GEOS-Chem (Goddard Earth Observing System chemical transport
model), Nowlan et al. (2014) applied the OMI $NO_2$ columns to calibrate the simulated ground $NO_2$
concentrations, and then estimated the deposition between 2005 and 2007. Our previous work focusing
on the dry particulate $NO_3^-$ deposition over China was also based on the OMI $NO_2$ columns, MOZART
simulations and monitored-based sources (Liu et al., 2017b). Geddes et al. (2017) used the satellite
$NO_2$ columns from GOME, GOME-2 and SCIAMACHY instruments to calibrate the $NO_x$ emissions in
GEOS-Chem to estimate the $NO_x$ depositions since 1996. The simulations combining the satellite
measurements and CTM models to derive Nr depositions (Geddes and Martin, 2017;Nowlan et al.,
2014) in recent years will provide relatively accurate datasets (certainly need to be validated and
modified by ground measurements).
Despite progress in satellite techniques in recent decades (for $NO_2$ since 1997 by GOME and for $NH_3$
since 2008 by IASI), we can hardly tracked studies concerning Nr depositions before 1997 based on
satellite observations. Thus, with the help of emissions data such as REAS and EDGAR, we can derive
long-term Nr depositions, especially before 1997. Long-term emissions data such as REAS and
EDGAR will provide valuable dataset to expand the modeling Nr depositions in recent years. In order
to derive the Nr depositions from the emission data, CTMs are frequently used through modeling the
wet (simplified as the product of scavenging efficiency and precipitation amount) and dry process
(simplified as the inferential method by multiplying the deposition velocity and gaseous or particulate
concentrations). However, we still lack a comprehensive dataset of gridded long-term Nr depositions
including both the dry ($NO_2$, $HNO_3$ particulate $NO_3^-$, $NH_3$ and particulate $NH_4^+$) and wet ($NH_4^+$ and
$NO_3^-$ in precipitation) processes over China, which will be addressed in future work.
Another gap is that, all the above mentioned studies focused on the $NO_x$ depositions and did not derive
the $NH_y$ ($NH_3$ and $NH_4^+$) depositions over China. Our recent work (Liu et al., 2017a) using IASI $NH_3$
columns combining the vertical profiles from MOZART benefits our understanding of the ground $NH_3$
concentrations over China, and the satellite-derived ground $NH_3$ concentrations were generally in
accord with the national measurements from NNDMN. To date, there are still no reports of using the
satellite $NH_3$ columns to derive the temporal and regional $NH_y$ depositions over China, which
dominated the total Nr depositions ($NO_x$ plus $NH_y$) (Liu et al., 2016b;Liu et al., 2013). The gaps of
modeling $NH_y$ depositions by applying the satellite observations combining the CTMs simulations
require more efforts and further research.

**4. Conclusion**

Atmospheric ammonia ($NH_3$) and nitrogen dioxide ($NO_2$) play an important role in determining air quality, environmental degradation and climate change. The emission data, satellite observations and atmospheric transport modeling have great potential for understanding the temporal variations of atmospheric $NH_3$ and $NO_2$ on a regional scale, with high spatial and temporal resolutions. This study analyzed the characteristics of atmospheric $NH_3$ and $NO_2$ over China since 1980 based on the multiple datasets. The major findings were as follows:

1. Based on emission data, both significant continuous increasing trend of $NH_3$ and $NO_x$ were observed from REAS (for $NH_3$ 0.17 kg N $ha^{-1}$ $y^{-2}$ and for $NO_x$ 0.16 kg N $ha^{-1}$ $y^{-2}$) and EDGAR (for $NH_3$ 0.24 kg N $ha^{-1}$ $y^{-2}$ and for $NO_x$ 0.17 kg N $ha^{-1}$ $y^{-2}$) over China during 1980-2010.

2. Based on the satellite observations, we found high-level $NH_3$ columns with the percent increase rate of 2.37% $y^{-1}$ from 2008 to 2014. For $NO_2$, we found continuous high-level $NO_2$ columns over China from 2005-2011 but a decrease from 2011 to 2015 (still in high level). The decrease of $NO_2$ columns may result from more stringent strategies taken to control $NO_x$ emissions during the 12th Five-Year-Plan, including successful application of SCR/SNCR systems in the power sector, tighter emission standards on vehicles and a series of standards with aggressive emission limits. Increasing trend of $NH_3$ columns may be due to continuous N fertilizer use for guaranteeing continuous increase of the crop productions. An increase in $NH_3$ columns may be due to decreased $NH_3$ removal leading to a larger fraction maintaining in gaseous state for a long time rather than changing to the condensed phase, which may be related with continuous decreased acidic gases including the $NO_2$ and $SO_2$ over China under strong control policy in 12-th FYP.

3. Based on MOZART simulations, the temporal variations of $NO_2$ columns at 12:00 from MOZART
were generally in accord with those from OMI at 13:45 P.M. local time. We also found a close increase
rate at 12:00 (4.02%) with that averaged at 00, 06, 12 and 18 h (4.23%) before 2011, as well as a
similar decrease rate at 12:00 (-2.93%) and the average (-3.07%). For $NH_3$, we found a lower percent
increase rate from MOZART (1.30% $y^{-1}$) than IASI (2.37% $y^{-1}$) between 2008 and 2014. Large
difference in the $NH_3$ increase rate in 2015 was found between IASI (50.45%) and MOZART (8.13%).
4. The multiple datasets used in the current work have implications for estimating long-term Nr
deposition datasets. The simulations combining the satellite measurements and CTM models to derive
Nr depositions will provide relatively accurate datasets, and the REAS and EDGAR emissions have
potential to expand the modeling Nr depositions to long-term datasets. In particular, modeling $NH_y$
depositions by applying the satellite observations combining the CTMs simulations require more
efforts and further research.
**Acknowledgements**
We acknowledge the free use of tropospheric $NO_2$ column data from the OMI sensor from
www.temis.nl. The $NH_3$ data have been obtained by the Atmospheric Spectroscopy Group at Université
Libre de Bruxelles (ULB) (http://www.ulb.ac.be/cpm/atmosphere.html). S. Whitburn and M. Van
Damme are acknowledged for making the data available and for their help in how to use them. We also
thank Louisa Emmons from National Center for Atmospheric Research (NCAR) for providing the
MOZART output data for the trend analysis. This study is supported by the National Natural Science
Foundation of China (No. 41471343, 40425007 and 41101315).

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

**Figures**

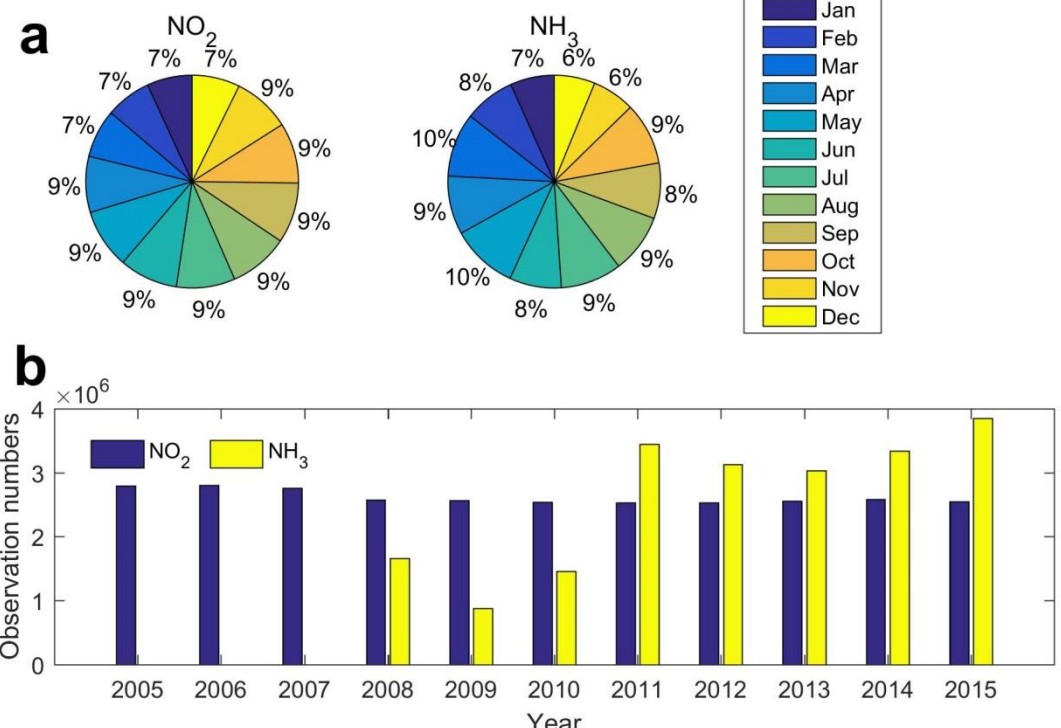


**Fig. 1**. The satellite-derived observation numbers for NO$_2$ and NH$_3$. (a) denotes the percentages of observations in each month in 2010 for NO$_2$ and in 2015 for NH$_3$ and (b) represents the total observation numbers for NO$_2$ and NH$_3$ over China. Notably, the NO$_2$ observation numbers were gained from DOMINO products with a cloud radiance fraction below 0.5, while the IASI observations with a relative error below 100% or an absolute error below $5\times10^{15}$ molec. cm$^{-2}$ were processed for analysis over China.

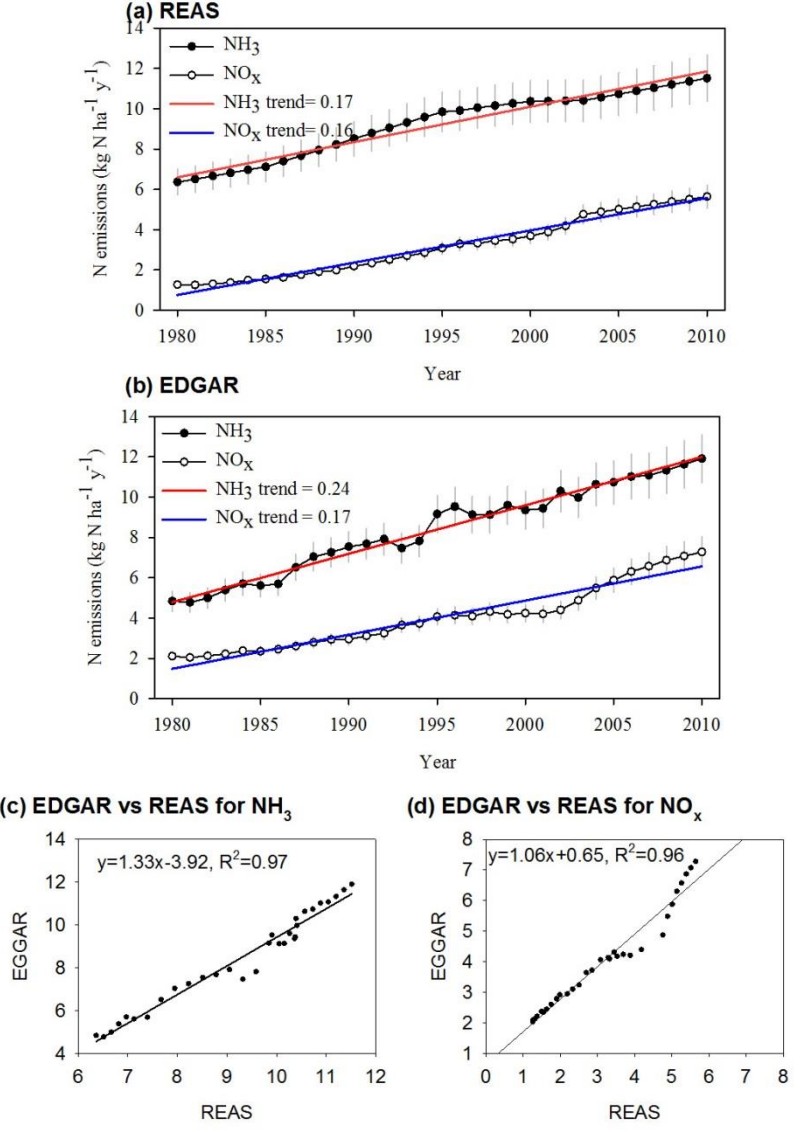


**Fig. 2**. The $NO_2$ and $NH_3$ emissions over China. (a) denotes the $NO_2$ and $NH_3$ emissions over China from 1980 to 2010 from
REAS, (b) represents the $NO_2$ and $NH_3$ emissions over China from 1980 to 2010 from EDGAR, (c) demonstrates the relationship
of $NO_2$ emissions over China from REAS and EDGAR and (d) shows the relationship of $NH_3$ emissions over China from REAS
and EDGAR.

**(a) OMI NO$_2$ at 13:45 P.M.**

slope (2011-2005)=0.063 y$^{-1}$ (**4.07% y$^{-1}$**)
slope (2015-2011)=0.072 y$^{-1}$ (**-3.62% y$^{-1}$**)

April-September
October-March

**(b) IASI NH$_3$ at 9:30 A.M.**

slope (2015-2008)=0.338 y$^{-1}$ (6.80% y$^{-1}$)
slope (2014-2008)=0.118 y$^{-1}$ (**2.37% y$^{-1}$**)
2015-2014: 50.45% y$^{-1}$

April-September
October-March


**Fig. 3.** Time series of average OMI NO$_2$ and IASI NH$_3$ columns over China during warm months (April-September) and cold
months (October-March). The time period of NO$_2$ columns was from 2005 to 2015, while the timespan of NH$_3$ columns was from
2008 to 2015 over China. The associated mean error for each period is presented here as error bars.


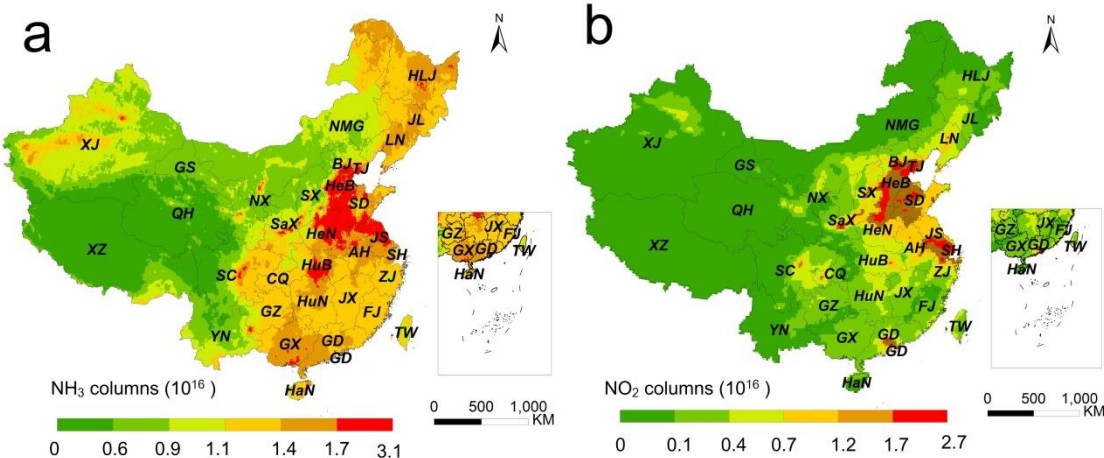


**Fig. 4.** Spatial distribution of the annual NH$_3$ (a) and NO$_2$ (b) columns (molecules cm$^{-2}$ year$^{-1}$). The successfully full provincial
names are Beijing (BJ), Tianjin (TJ), Hebei (HeB), Shandong (SD), Shanxi (SX), Henan (HeN), Shaanxi (SaX), Liaoning (LN),
Jilin (JL), Heilongjiang (HLJ), Neimenggu (NMG), Gansu (GS), Ningxia (NX), Xinjiang (XJ), Shanghai (SH), Jiangsu (JS),
Zhejiang (ZJ), Anhui (AH), Hubei (HuB), Hunan (HuN), Jiangxi (JX), Fujian (FJ), Guangdong (GD), Hainan (HaN), Yunnan
(YN), Guizhou (GZ), Chongqing (CQ), Sichuan (SC), Guangxi (GX), Xizang (XZ) and Qinghai (QH).

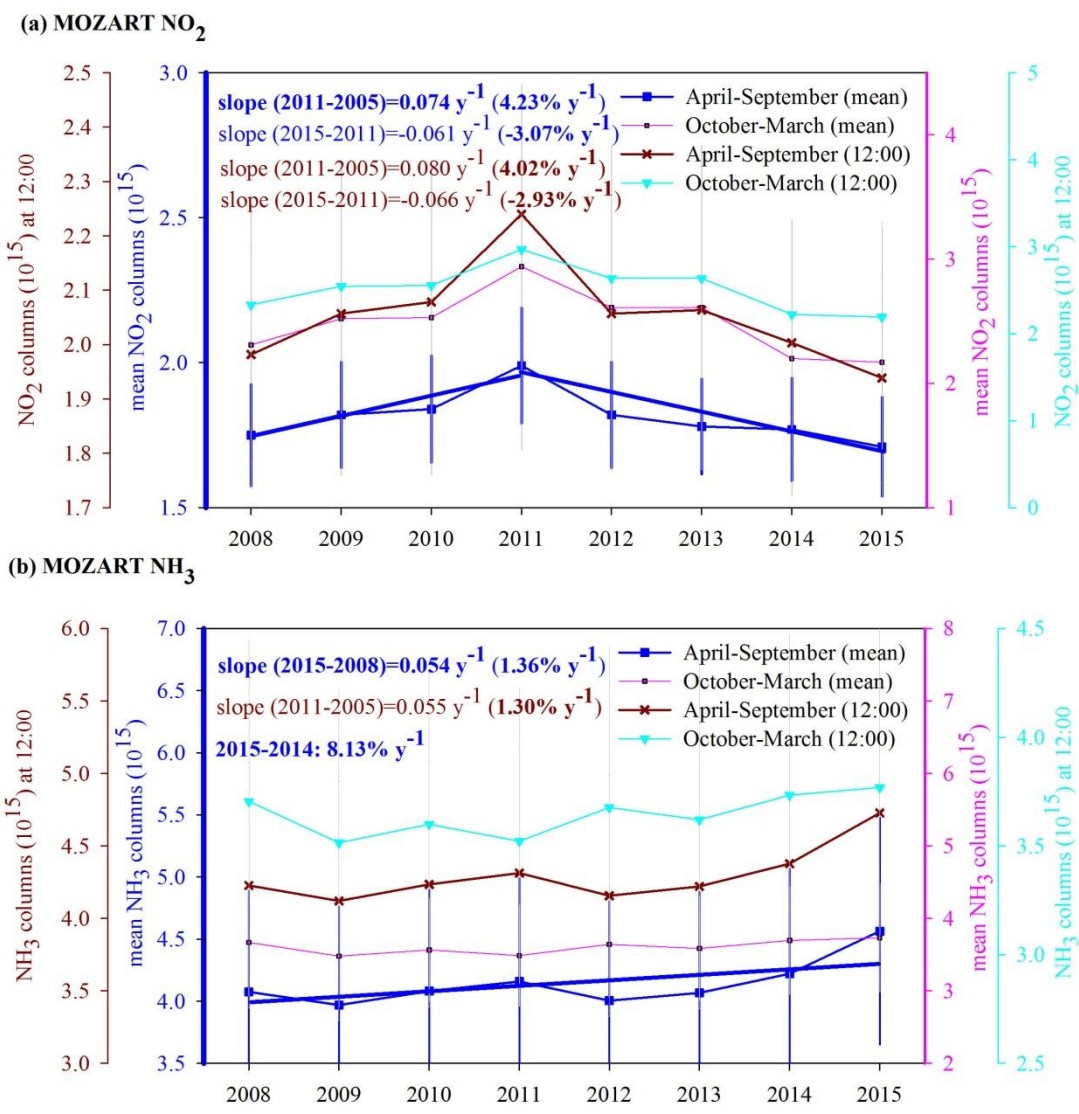

**Fig. 5.** Time series of MOZART $NO_2$ and $NH_3$ columns over China during average warm months (April-September) and cold
months (October-March) from 2008 to 2015. The mean columns were calculated by averaging the columns at 00, 6, 12 and 18 h.
The associated mean error for each period is presented here as error bars.


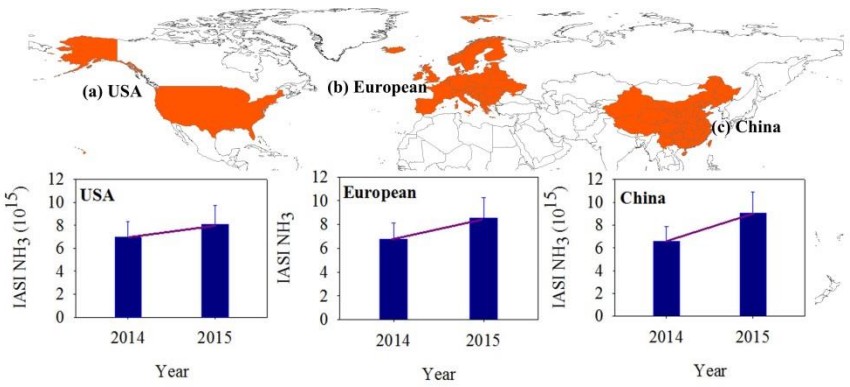

**Fig. 6**. IASI NH$_3$ columns in USA, European and China between 2014 and 2015.