# Peer review of "Temporal characteristics of atmospheric ammonia and nitrogen dioxide over China based on"

_Atmospheric Chemistry and Physics, 2017_

## Referee Comment (RC1) · Anonymous Referee #1 · 14 Apr 2017

This manuscript presents an overview of the temporal characteristics of various datasets relevant to NOx and NH3 abundance over China. The authors discuss trends in emission inventories (EDGAR and REAS), trends in satellite NO2 and NH3 columns (from OMI and IASI respectively), and trends in MOZART-4 model output for the region. Decreasing NO2 since 2011 suggests that China's 12th Five Year Plan has resulted in successful emission reductions. On the other hand, the lack of a significant trend in NH3 points to the growing importance of controlling and monitoring reduced nitrogen.

The authors are to be commended for compiling and exploring multiple datasets in

determining patterns in reactive nitrogen over China. However, I have some general comments about the overall scientific significance and scientific quality. I look forward to hearing from the authors in this discussion phase.

(1) While the analysis of IASI NH3 columns focusing on China might be somewhat new, I find the analysis of OMI NO2 that is presented in this manuscript lacking in novelty or insight. In particular, I would refer the authors to de Foy et al. (2016) and to Liu et al. (2016). Both of these studies use OMI NO2 observations from 2005-2015 to discuss long-term trends and the 2011 peak in NO2 over China in detail. In my opinion, the observations made by the authors of this present manuscript have not added new insight into this discussion (and in fact treat the analysis with less rigor, as I will discuss below). In its current state, I am concerned that this manuscript does not represent a substantial enough contribution. I encourage the authors to refer to the above references and explicitly address what new insight is gained from their analysis.

(2) The inclusion of model results has added very little insight to the analysis. The MOZART model is driven by the EDGAR emissions to begin with (which are discussed in more detail separately). For both NO2 and NH3, I would expect the relationship between emissions and tropospheric columns to be pretty strong, so it's not clear what is expected to be learned by comparing trends in EDGAR emissions with trends in model output based on EDGAR emissions. Moreover, there is no analysis or discussion of the NO2 model output at all, so why has this output been included in the figures? The authors must expand on or address why the model output has been included, and demonstrate clearly what insight is gained.

(3) The authors conclude the discussion section with implications for estimating long-term reactive nitrogen deposition. The discussion about uncertainty and challenges in estimating dry and wet deposition seems to be out of place in this manuscript. Of course, there is an obvious connection between emissions, atmospheric abundance, and deposition – but this manuscript does not bring up the question of deposition until this final section, so it appears as a digression. While I agree with the conclusion

made by the authors (that more long-term data sets are needed), I feel their discussion has not presented any new concepts based specifically on the results presented in this manuscript. The connection between their analysis and insight into nitrogen deposition should be made stronger throughout the manuscript. Specifically, what has been gained from the analysis?

Technical Comment:

(1) In determining the trend in NO2, the authors have calculated a linear fit to the monthly average data. I find this approach to be problematic, since the trend seems to be influenced strongly by an increasing seasonal amplitude. In my opinion, the authors need to remove (or account for) the seasonality before calculating a long-term trend. Specifically, the winter monthly means seem to be driving most of the increase in their linear fit – but these values have the highest uncertainty (borne out by the larger magnitude of the error bars compared to summer months). Accounting for seasonality in determining trends in NO2 is common practice. This can be accomplished, for example, by fitting the seasonal amplitude separately (e.g. Lamsal et al. (2015)), or by calculating trends in seasonal averages (e.g. Russel et al. (2012)).

Specific comments:

line 88: The authors use of the term "widely" warrants more than two examples in the citation.

line 110: "is believed to have the highest spatial resolution". Surely this statement can be confirmed instead of believed.

line 117: I suggest the authors replace the expression "multivariate", since this term usually implies something different (i.e. modeling). May I suggest the authors use "multiple datasets" throughout the manuscript, instead of "multivariate".

lines 151-153: Repeating the thresholds for error consideration is redundant here.

line 202: Please also include the spatial resolution of the model simulation.

line 223: "their thread values both positive". Please clarify this sentence.

line 232-233: I think the closer agreement with one other estimate does not necessarily mean the EDGAR estimate is "more reasonable". Please qualify.

line 255 (and elsewhere): The use of the expression "no big changes" does not have much scientific meaning. May I suggest "no significant changes" followed by the results of some statistical test?

line 256: The slope in NH3 of 0.025 x 10ˆ15 is actually twice the slope of NO2 (0.011 x 10ˆ15), so can the authors clarify why the slope in NH3 is not determined to be important or large? Should they clarify that they are speaking in relative terms to the atmospheric concentrations?  What are the trends in %/year for NO2 compared to NH3?

line 305-306: Can the author confirm these numbers are coming from the reference in the preceding sentence (Wang et al. 2012)?

line 311: Can the authors explain why it would be better to calculate trends based on daily data? This would be unusual.

line 350, 351, and 353: Are the authors referring to the panels in Figure 6, not Figure 5?

line 359: "...this is the conclusion we really concerned." Please clarify this sentence.

line 360: "... the following discussion in this paragraph was all hypothetical". Are the authors referring to the next two sentences? This isn't much of a discussion.

line 373: "in high level". I suggest replacing this expression with something more clear.

line 401: "no big variations". Again, I suggest replacing this statement with something more scientifically/statistically clear.

References:

[Figure]

de Foy et al. (2016), Scientific Reports, http://dx.doi.org/10.1038/srep35912

Liu et al. (2016), Environmental Research Letters, http://dx.doi.org/10.1088/1748-9326/11/11/114002

Lamsal et al. (2015), Atmospheric Environment, http://dx.doi.org/10.1016/j.atmosenv.2015.03.055

Russell et al. (2012), Atmospheric Chemistry and Physics, http://dx.doi.org/10.5194/acp-12-12197-2012
* * *

---

## Short Comment (SC1) · 21 Apr 2017

Martin Van Damme (1), Lieven Clarisse (1), Simon Whitburn (1), Enrico Dammers (2), Pierre-François Coheur (1)

(1) Université libre de Bruxelles (ULB), Atmospheric Spectroscopy, Service de Chimie Quantique et Photophysique CP 160/09, av. F.D. Rossevelt 50, 1050 Bruxelles, Belgium

(2) Cluster Earth and Climate, Department of Earth Sciences, Vrije Universiteit Amsterdam, Amsterdam, the Netherlands

This manuscript presents time-series of NH3 total columns from the IASI satellite instrument. As the authors point out, the retrieved columns for this trace gas are strongly dependent on the meteorological information that is used as input data in the retrieval algorithm (in particular the surface temperature and the temperature profile in the lowest layers of the atmosphere). At present, the publically available IASI-NH3 product uses the meteorological parameters as provided in EUMETSAT IASI Level 2 data.

The IASI Level 2 data is processed in real time, and there have been several updates to the algorithm since the launch of IASI. We can confirm that in particular, the change from version 5 to version 6 of the EUMETSAT Level 2 data on 30 September 2014 (and not November as stated in the manuscript), causes on average a substantial increase of the retrieved atmospheric NH3 columns. Hence the conclusions of the authors regarding NH3 trends should be taken with great care.

---

## Referee Comment (RC2) · Anonymous Referee #2 · 30 Apr 2017

Review of "Temporal characteristics of atmospheric ammonia and nitrogen dioxide over China based on emission data, satellite observations and atmospheric transport modeling since 1980" by Lei Liu et al.

This manuscript showed interesting results on the temporal evolution of $NO_x$ and $NH_3$ over China. By comparing the data resulting from inventories of REAS and EDGAR, the authors found that $NH_3$ and $NO_x$ continually increased over China during 1980-2010. Furthermore, based on previous satellite observations and an atmospheric chemistry transport model (MOZART-4), they also found that $NO_2$ over China increased from 2005 to 2011 and then decreased significantly from 2011 to 2015. Finally the authors discussed the plausible reasons including control policies of Chinese government to the emission trends of reactive nitrogen. Overall the topic of the study is sound and the manuscript was written well. However, I have the following concerns to be addressed before recommending it for publication in *Atmos. Chem. Phys.*

Major comments:

1. In line 168 of page 8, the authors filtered the DOMINO product with an absolute error below $10^{15}$ molecules $cm^{-2}$. However the $NO_2$ vertical column densities (VCDs) error depend on the net values of $NO_2$ VCDs. Therefore the filter may arbitrarily exclude the high $NO_2$ VCD values. The authors should evaluate the influence of absolute errors on the final emission results and show it in current study.

2. The authors compared the emission data of $NO_2$ and $NH_3$ from satellite observations to that from Mozart-4 model simulations. But the authors did not explain whether the satellite overpass time has been considered during the comparison or not. The OMI satellite only gives the $NO_2$ data at about 1:30 pm of local time. The same time could also be used for the extraction of $NO_2$ data from Mozart-4 model. Whether this will influence the output results and conclusions of current study? This point should be clarified more.

3. The MOZART-4 model contained 12 bulk aerosol compounds, 39 photolysis, 85 gas species as well as 157 gas-phase reactions. However, the authors did not discuss the influence of $NO_x$ and $NH_3$ sink on their emission values at all while elucidating the data from MOZART-4. Although the authors have discussed the potential impacts of emission regulation or

energy efficiency enhancement relevant government control policies on the $NO_x$ and $NH_3$ emissions, they are encouraged to show their insight on the correlations of atmospheric process of $NO_x$ and $NH_3$ with their final emission values.

4. In section 3.1, the authors showed the emission data result from REAS and EDGAR, but they did not give convincing reasons for the different results of 0.24 kg N $ha^{-1}$ $y^{-2}$ from EDGAR and 0.17 kg N $ha^{-1}$ $y^{-2}$ from REAS. The authors should supply plausible explanations (e.g. induced by methodological difference of data compiling or meteorological factors etc.) to this. In addition, the authors thought 0.24 kg N $ha^{-1}$ $y^{-2}$ from EDGAR was much higher than 0.17 kg N $ha^{-1}$ $y^{-2}$ from REAS in lines 221-222 of page 11. However, they thought 0.33 kg N $ha^{-1}$ $y^{-2}$ was close to 0.24 kg N $ha^{-1}$ $y^{-2}$ in lines 231-232 of the same page. This is logically wrong. They need to correct it and also the relevant discussions.

5. In lines 311-315 of page 15, the whole daily coverage over China cannot be achieved also due to the row anomaly effect. This effect may cause half of the satellite pixels to be unusable. The discussions here should be rearranged.

6. Lines 99-101: the authors are encouraged to expand introduction on the method for converting satellite data to $NH_3$ column. Only a reference citation is not convenient for readers to follow up the work in a straight way.

Minor comments:

7. Line 102: the words of 'provides' and 'potential' should be changed to 'provide' and 'possibility'.

8. Line 104: the description of 'emission data are also very important tools' is confusing, and there is no logic comparability with 'satellite observations' in the front dialogue, so I suggest to remove the 'tools' or modify the front dialogue properly.

9. Line 110: change 'resolutions' to 'resolution'.

10. Line 170: change 'the manuscript' to 'previous work'.

11. Line 130: change 'denotes' to 'denote'.

12. Line 228-29: Similar information of the first dialogue here has been shown in lines 221-222, so there is no necessary to show it twice.

13. Line 229-230: the description of 'Liu et al. (2013) conducted that emissions of national anthropogenic $NH_3$ and $NO_x$ summarized from published data during 1980-2010' is confusing and should be rearranged.

14. Figure 1: add error bars to panel b please.

---

## Author Comment (AC1) · 16 Jun 2017

**Referee #1**

We are grateful to the reviewer for the time and energy in providing helpful comments and guidance that have improved the manuscript. In this document, we describe how we have addressed the reviewer's comments. Detailed responses to each comment are given below (in blue).

This manuscript presents an overview of the temporal characteristics of various datasets relevant to NOx and $NH_3$ abundance over China. The authors discuss trends in emission inventories (EDGAR and REAS), trends in satellite $NO_2$ and $NH_3$ columns (from OMI and IASI respectively), and trends in MOZART-4 model output for the region. Decreasing $NO_2$ since 2011 suggests that China's 12th Five Year Plan has resulted in successful emission reductions. On the other hand, the lack of a significant trend in $NH_3$ points to the growing importance of controlling and monitoring reduced nitrogen. The authors are to be commended for compiling and exploring multiple datasets in deter mining patterns in reactive nitrogen over China. However, I have some general comments about the overall scientific significance and scientific quality. I look forward to hearing from the authors in this discussion phase.

**Major comments:**

(1) While the analysis of IASI $NH_3$ columns focusing on China might be somewhat new, I find the analysis of OMI $NO_2$ that is presented in this manuscript lacking in novelty or insight. In particular, I would refer the authors to de Foy et al. (2016) and to Liu et al. (2016). Both of these studies use OMI $NO_2$ observations from 2005-2015 to discuss long-term trends and the 2011 peak in $NO_2$ over China in detail. In my opinion, the observations made by the authors of this present manuscript have not added new insight into this discussion (and in fact treat the analysis with less rigor, as I will discuss below). In its current state, I am concerned that this manuscript does not represent a substantial enough contribution. I encourage the authors to refer to the above references and explicitly address what new

insight is gained from their analysis.

The new insights gained from this study are for Ammonia ($NH_3$) as well as the potential interactive impact between $NO_2$ and $NH_3$. The temporal trend analysis of $NH_3$ columns over China in the present work is relatively new, and to date studies focusing on the $NH_3$ trends based on the IASI observations over China are still few.

Although there have been several studies regarding the temporal trends of $NO_2$ columns over China including Foy et al. (2016) and Liu et al. (2016), their analysis did not show the discussion on the possible interactions between $NO_2$ and $NH_3$. $NH_3$ is the most abundant alkaline gas in the troposphere and is important for its ability to neutralize acidic components such as sulfuric acid ($H_2SO_4$) and nitric acid ($HNO_3$) which form, respectively, by the oxidation of emissions of sulfur dioxide ($SO_2$) and nitrogen oxides ($NO_x$). Reactions of $HNO_3$ and $H_2SO_4$ with $NH_3$ generally form submicron ammonium nitrate ($NH_4NO_3$) and ammoniated sulfate ($NH_4HSO_4$, $(NH_4)_2SO_4$, or other forms) particles. High temperatures also promote dissociation of $NH_4NO_3$ back to gaseous $NH_3$ and $HNO_3$. Therefore, the temporal trends of $NH_3$ and $NO_2$ should have an interactive impact between each other.

An increase in $NH_3$ columns in recent years may also be due to decreased $NH_3$ removal leading to a larger fraction remaining in a gaseous state for a long time rather than changing to the condensed phase, which can be attributed to continuous decreased acidic gases over China including the $NO_2$ and $SO_2$ under strong control policy in 12-th FYP. This can largely decrease the fraction of the chemical conversion to $(NH_4)_2SO_4$ and $NH_4NO_3$ in the atmosphere (Paragraph 2 in Sect. 3.2.1).

In addition, we used different methods than Foy et al. (2016) and Liu et al. (2016). We adopted the method of Russel et al. (2012) (concentrating on the US in the original paper) to quantify the change of $NO_2$ columns over China with focusing on the temporal analysis of warm months due to the relatively

Technical Comment:

(1) In determining the trend in $NO_2$, the authors have calculated a linear fit to the monthly average data.

I find this approach to be problematic, since the trend seems to be influenced strongly by an increasing

seasonal amplitude. In my opinion, the authors need to remove (or account for) the seasonality before

calculating a long-term trend. Specifically, the winter monthly means seem to be driving most of the

increase in their linear fit – but these values have the highest uncertainty (borne out by the larger

magnitude of the error bars compared to summer months). Accounting for seasonality in determining

trends in $NO_2$ is common practice. This can be accomplished, for example, by fitting the seasonal

amplitude separately (e.g. Lamsal et al. (2015)), or by calculating trends in seasonal averages (e.g.

Russel et al. (2012)).

Here, we respond to the Technical Comment before other major comments.

We agree with the reviewer that considering seasonality in determining trends of $NO_2$ is important. We

have carefully reviewed all the given references including Foy et al. (2016) and Liu et al. (2016) in

Major comments (1) as well as the given references of Lamsal et al. (2015) and Russell et al. (2012) in

the Technical Comment.

We adopted the method of Russel et al. (2012) as suggested by the reviewer. In this method, averages

were computed for both cold months (October-March) and warm months (April-September). We

concentrated more on the temporal analysis of warm months due to the relatively low uncertainty

compared to cold months. We have added related explanations and introduction text at Paragraph 1 in

Sect. 3.2.1.

[Figure]

**Fig. 3.** Time series of average OMI $NO_2$ and IASI $NH_3$ columns over China during warm months (April-September) and cold months (October-March). The time period of $NO_2$ columns was from 2005 to 2015, while the timespan of $NH_3$ columns was from 2008 to 2015 over China. The associated mean error for each period is presented here as error bars.

(2) The inclusion of model results has added very little insight to the analysis. The MOZART model is

driven by the EDGAR emissions to begin with (which are discussed in more detail separately). For

both $NO_2$ and $NH_3$, I would expect the relationship between emissions and tropospheric columns to be pretty strong, so it's not clear what is expected to be learned by comparing trends in EDGAR emissions with trends in model output based on EDGAR emissions. Moreover, there is no analysis or discussion of the $NO_2$ model output at all, so why has this output been included in the figures? The authors must expand on or address why the model output has been included, and demonstrate clearly what insight is gained.

Satellite $NO_2$ and $NH_3$ columns were observed at overpass time as an instantaneous point in a day (at 9:30 A.M. for IASI $NH_3$ and at 1:45 P.M. for OMI $NO_2$ local time). These instantaneous satellite observations may not be representative for the temporal trend analysis over China (refer to Paragraph 1 in Sect. 3.3). We calculated the monthly mean $NO_2$ and $NH_3$ columns from MOZART varying 6 hours every day (00, 06, 12, 18 h) in order to: (1) calculate the temporal trend of mean $NO_2$ and $NH_3$ columns (averaged at 00, 06, 12, 18 h) rather than the instantaneous values; (2) compare the temporal trend analysis of $NO_2$ from MOZART at 12 h with that gained from satellite at the overpass time (OMI 1:45 P.M. local time) as well as for $NH_3$.

In general, we found an agreement on the $NO_2$ temporal trend between MOZART (12:00) and OMI (13:45), while we found a lower increase rate from MOZART (12:00) than from IASI (9.30 A.M.). We have expanded the analysis and discussion of the $NO_2$ as well as $NH_3$ from MOZART at Paragraph 2 and 3 in Sect. 3.3. Please refer to them.

[Figure]

**Fig. 5.** Time series of MOZART NO$_2$ and NH$_3$ columns over China during average warm months (April-September) and cold months (October-March) from 2008 to 2015. The mean columns were calculated by averaging the columns at 00, 6, 12 and 18 h. The associated mean error for each period is presented here as error bar.

(3) The authors conclude the discussion section with implications for estimating long term reactive nitrogen deposition. The discussion about uncertainty and challenges in estimating dry and wet deposition seems to be out of place in this manuscript. Of course, there is an obvious connection between emissions, atmospheric abundance, and deposition - but this manuscript does not bring up the question of deposition until this final section, so it appears as a digression. While I agree with the conclusion made by the authors (that more long-term data sets are needed), I feel their discussion has not presented any new concepts based specifically on the results presented in this manuscript. The

connection between their analysis and insight into nitrogen deposition should be made stronger throughout the manuscript. Specifically, what has been gained from the analysis?

We agree with you that the connection between the trend analysis and insight into nitrogen deposition should be made stronger. The current paper describes the temporal characteristics of atmospheric $NH_3$ and $NO_2$ over China based on multiple datasets including emission data, satellite observations and atmospheric transport modeling results since 1980. We believe the multiple datasets used in the current work have implications for estimating long-term reactive nitrogen (Nr) deposition datasets, and Sect. 3.4 describes this point and the future work will be done soon. We have changed the title from "3.4. Implications for estimating long-term Nr deposition datasets" to "3.4. Implications for estimating long-term Nr deposition datasets and recommendations for future work". To make this point more clear, we have added the following text for more clarification:

"Satellite observation provides a new perspective for estimating Nr depositions regionally. For example, to improve the modeling performance in dry gaseous $NO_2$ depositions from GEOS-Chem (Goddard Earth Observing System chemical transport model), Nowlan et al. (2014) applied the OMI $NO_2$ columns to calibrate the simulated ground $NO_2$ concentrations, and then estimated the deposition between 2005 and 2007. Our previous work focusing on the dry particulate $NO_3^-$ deposition over China was also based on the OMI $NO_2$ columns, MOZART simulations and monitored-based sources (Liu et al., 2017b). Geddes et al. (2017) also used the satellite $NO_2$ columns from GOME, GOME-2 and SCIAMACHY instruments to calibrate the $NO_x$ emissions in GEOS-Chem to estimate the $NO_x$ depositions since 1996. The simulations combining the satellite measurements and CTM model to derive Nr depositions (Geddes and Martin, 2017;Nowlan et al., 2014) in recent years will provide relatively accurate datasets (certainly need to be validated and modified by ground measurements)."

"Despite progress in satellite techniques in recent decades (for $NO_2$ since 1997 by GOME and for $NH_3$ since 2008 by IASI), we can hardly tracked studies concerning Nr depositions before 1997 based on satellite observations. Thus, with the help of emissions data such as REAS and EDGAR, we can derive long-term Nr depositions, especially before 1997. Long-term emissions data such as REAS and EDGAR will also provide a valuable dataset to expand the modeling Nr depositions in recent years. In order to derive the Nr depositions from the emission data, the atmospheric chemistry transport models (CTMs) are frequently used through modeling the wet (simplified as the product of scavenging efficiency and precipitation amount) and dry processes (simplified as the inferential method by multiplying the deposition velocity and gaseous or particulate concentrations). However, we still lack a comprehensive dataset of gridded long-term Nr depositions including both the dry ($NO_2$, $HNO_3$, particulate $NO_3^-$, $NH_3$ and particulate $NH_4^+$) and wet ($NH_4^+$ and $NO_3^-$ in precipitation) processes over China, which will be addressed in future work".

"Another gap is that, all the above mentioned studies focused on the $NO_x$ depositions and did not derive the $NH_y$ ($NH_3$ plus $NH_4^+$) depositions over China. Our recent work (Liu et al., 2017a) using IASI $NH_3$ columns combining the vertical profiles from MOZART benefits our understanding of the ground $NH_3$ concentrations over China, and the satellite-derived ground $NH_3$ concentrations were generally in accord with the national measurements from NNDMN. To date, there are still no reports of using the satellite $NH_3$ columns to derive the temporal and regional $NH_y$ depositions over China, which dominated the total Nr depositions ($NO_x$ plus $NH_y$) (Liu et al., 2016;Liu et al., 2013). The gaps of modeling $NH_y$ depositions by applying the satellite observations combining the CTMs simulations require more efforts and further research".

We herein list some important works regarding Nr depositions using satellite, CTMs and emissions as

well as cited them in the main text:

(1) Liu, L., Zhang, X., Zhang, Y., Xu, W., Liu, X., Zhang, X., Feng, J., Chen, X., Zhang, Y., Lu, X., Wang, S., Zhang, W., and Zhao, L.: Dry Particulate Nitrate Deposition in China, Environmental Science & Technology, 10.1021/acs.est.7b00898, 2017. **Our recent work focused on the dry $NO_3^-$ deposition based on OMI $NO_2$, MOZART simulations and monitor-based sources**.

(2) Liu, L., Zhang, X., Xu, W., Liu, X., Lu, X., Wang, S., Zhang, W., and Zhao, L.: Ground Ammonia Concentrations over China Derived from Satellite and Atmospheric Transport Modeling, Remote Sensing, 9, 467, 2017. **Our recent work focused on ground $NH_3$ concentrations based on IASI $NH_3$ and MOZART simulations, and we can gain dry $NH_3$ depositions combining the deposition velocity**.

(3) Zhang, X., Lu, X., Liu, L., Chen, D., Zhang, X., Liu, X., Zhang, Y.: Dry gaseous $NO_2$ deposition inferred from Ozone Monitoring Instrument $NO_2$ columns and atmospheric chemistry transport model over China, Journal of Geophysical Research-Atmosphere, 2017 (submitted). **Our recent work focused on the gaseous $NO_2$ depositions based on OMI $NO_2$ and MOZART simulations**.

(4) Geddes, J. A., and Martin, R. V.: Global deposition of total reactive nitrogen oxides from 1996 to 2014 constrained with satellite observations of NO2 columns, Atmos. Chem. Phys. Discuss., 2017, 1-44, 10.5194/acp-2016-1100, 2017. **Geddes's recent work focused on the total $NO_x$ depositions globally based on the GOME, GOME-2 and SCIAMACHY $NO_2$ and GEOS-Chem.**

(5) Nowlan, C., Martin, R., Philip, S., Lamsal, L., Krotkov, N., Marais, E., Wang, S., and Zhang, Q.: Global dry deposition of nitrogen dioxide and sulfur dioxide inferred from space‐based measurements, Global Biogeochemical Cycles, 28, 1025-1043, 2014. **Nowlan's previous work focused on the gaseous $NO_2$ depositions globally based on the OMI $NO_2$ and GEOS-Chem.**

Specific comments:

line 88: The authors use of the term "widely" warrants more than two examples in the citation.

We added 4 new references in this line: Castellanos et al., 2015, Lamsal et al., 2015, Liu et al., 2016 and Foy et al., 2016.

line 110: "is believed to have the highest spatial resolution". Surely this statement can be confirmed instead of believed.

We have changed "is believed to have the highest spatial resolution" to "has the highest spatial resolution".

line 117: I suggest the authors replace the expression "multivariate", since this term usually implies something different (i.e. modeling). May I suggest the authors use "multiple datasets" throughout the manuscript, instead of "multivariate".

We have changed "multivariate data" to "multiple datasets" throughout the manuscript.

lines 151-153: Repeating the thresholds for error consideration is redundant here.

We have removed the repetition in these lines.

line 202: Please also include the spatial resolution of the model simulation.

We have added it as suggested.

line 223: "their thread values both positive". Please clarify this sentence.

We have changed "their thread values both positive" to "their thread values of 0.24 kg N ha$^{-1}$ y$^{-2}$ (EDGAR) vs 0.17 kg N ha$^{-1}$ y$^{-2}$ (REAS) both reflected a continuous increasing trend (in this regard they are consistent)".

line 232-233: I think the closer agreement with one other estimate does not necessarily mean the EDGAR estimate is "more reasonable". Please qualify.

The original discussion in line 232-233 was logically wrong, and we are now aware of that. Reviewer 2 also commented, "the authors thought 0.24 kg N ha$^{-1}$ y$^{-2}$ from EDGAR was much higher than 0.17 kg N ha$^{-1}$ y$^{-2}$ from REAS in lines 221-222 of page 11. However, they thought 0.33 kg N ha$^{-1}$ y$^{-2}$ was close to 0.24 kg N ha$^{-1}$ y$^{-2}$ in lines 231-232 of the same page. This is logically wrong. They need to correct it and also the relevant discussions."

In this revision, we have rewritten the sentences as the following text in the third paragraph in Sect. 3.1:

"A previous study (Liu et al., 2013) summarized published data on the national anthropogenic NH$_3$ and NO$_x$ emissions with multi-periods in China (Wang et al., 2009;Wang et al., 1997;Streets et al., 2003;Klimont et al., 2001;Sun and Wang, 1997;Olivier et al., 1998;FRCGC, 2007), and also analyzed the temporal pattern of NH$_3$ emissions. Their results showed that the NH$_3$ emissions had increased at an annual average rate of 0.32 Tg N y$^{-2}$ (about 0.33 kg N ha$^{-1}$ y$^{-2}$). The increase rate of NH$_3$ emissions (0.33 kg N ha$^{-1}$ y$^{-2}$) by Liu et al. (2013) was double that in REAS (0.17 kg N ha$^{-1}$ y$^{-2}$), implying that the NH$_3$ increase rate in China is still an open question, and should be further studied in future work.".

line 255 (and elsewhere): The use of the expression "no big changes" does not have much scientific meaning. May I suggest "no significant changes" followed by the results of some statistical test?

We have changed "no big changes" to "$0.118 \times 10^{15}$ molec. cm$^{-2}$ y$^{-1}$ (2.37% y$^{-1}$) in warm months".

line 256: The slope in NH3 of 0.025 x 10^15 is actually twice the slope of NO2 (0.011 x 10^15), so can the authors clarify why the slope in NH3 is not determined to be important or large? Should they clarify that they are speaking in relative terms to the atmospheric concentrations? What are the trends in %/year for NO2 compared to NH3?

Yes, we refer to the percent increase rate rather than the absolute increase rate. We have added the

following text for explanations as well as the percent increase rate (% $y^{-1}$) by the following text:

"The percent increase rate for $NH_3$ by year (2.37% $y^{-1}$) from 2008 to 2014 is lower than that for $NO_2$ (4.07% $y^{-1}$) from 2005 to 2011, although the absolute $NH_3$ increase rate of $0.118 \times 10^{15}$ molec. $cm^{-2}$ $y^{-1}$ from 2008 to 2014 was higher than that of $0.063 \times 10^{15}$ molec. $cm^{-2}$ $y^{-1}$ for $NO_2$ from 2005 to 2011.".

line 305-306: Can the author confirm these numbers are coming from the reference in the preceding sentence (Wang et al. 2012)?

No, these numbers come from the reference (Xia et al., 2016), and we have added the reference.

line 311: Can the authors explain why it would be better to calculate trends based on daily data? This would be unusual.

This sentence has been rewritten and clarified by the following text:

"It is difficult to gain whole coverage based on the daily data over China for both IASI $NH_3$ and OMI $NO_2$. For daily $NO_2$, the spatial coverage gained by OMI were influenced by cloud radiance fractions, surface albedo, solar zenith angles, row anomaly and so on (Russell et al., 2011;De Smedt et al., 2015). "row anomaly" issue resulting from the OMI instrumental problem had an impact on approximately half of the rows undergoing unpredictable patterns in cross-track directions relying on latitudes and seasons and prevented obtaining convincing daily product with continuous coverage (Boersma et al., 2011;Boersma et al., 2016).".

line 350, 351, and 353: Are the authors referring to the panels in Figure 6, not Figure 5?

Yes, we have changed it.

line 359: "...this is the conclusion we really concerned." Please clarify this sentence.

We referred to the sentence "At the current state, we can, at least, draw a conclusion that the $NH_3$ columns over China indeed increased in 2015 both from IASI and MOZART, but a debate or

inconsistency exists on the increase rate of the $NH_3$ columns in 2015". We have marked this sentence in red and removed "this is the conclusion we really concerned".

line 360: "... the following discussion in this paragraph was all hypothetical". Are the authors referring to the next two sentences? This isn't much of a discussion.

Yes, we refer to the sentence: "For IASI $NH_3$ columns, the sharp increase in 2015 over China may be an artifact, which may be due to an update of the input data."

line 373: "in high level". I suggest replacing this expression with something more clear.

We have changed "in high level" to "in high level with an average of 1.87 molec. $cm^{-2}$ $y^{-1}$ compared with that (1.65 molec. $cm^{-2}$ $y^{-1}$) during 2005-2010".

line 401: "no big variations". Again, I suggest replacing this statement with something more scientifically/statistically clear.

We have changed "no big variations" to "the percent increase rate of 2.37% $y^{-1}$".

References:

de Foy et al. (2016), Scientific Reports, http://dx.doi.org/10.1038/srep35912

Liu et al. (2016), Environmental Research Letters, http://dx.doi.org/10.1088/1748-9326/11/11/114002

Lamsal et al. (2015), Atmospheric Environment, http://dx.doi.org/10.1016/j.atmosenv.2015.03.055

Russell et al. (2012), Atmospheric Chemistry and Physics, http://dx.doi.org/10.5194/acp-12-12197-2012

We have reviewed and added all the suggested references.

**Other corrections**

Removed original Fig. 6.

Since the information on the increase rate (%) between 2014 and 2015 from MOZART and IASI has

been added in Fig. 3 and Fig. 5 in this revision, we have removed original Fig. 6 to avoid duplication.

**References**

Boersma, K. F., Eskes, H. J., Dirksen, R. J., van der A, R. J., Veefkind, J. P., Stammes, P., Huijnen, V., Kleipool, Q. L., Sneep, M., Claas, J., Leitão, J., Richter, A., Zhou, Y., and Brunner, D.: An improved tropospheric NO2 column retrieval algorithm for the Ozone Monitoring Instrument, Atmospheric Measurement Techniques, 4, 1905-1928, 10.5194/amt-4-1905-2011, 2011.

Boersma, K. F., Vinken, G. C. M., and Eskes, H. J.: Representativeness errors in comparing chemistry transport and chemistry climate models with satellite UV–Vis tropospheric column retrievals, Geosci. Model Dev., 9, 875-898, 10.5194/gmd-9-875-2016, 2016.

De Smedt, I., Stavrakou, T., Hendrick, F., Danckaert, T., Vlemmix, T., Pinardi, G., Theys, N., Lerot, C., Gielen, C., and Vigouroux, C.: Diurnal, seasonal and long-term variations of global formaldehyde columns inferred from combined OMI and GOME-2 observations, Atmospheric Chemistry & Physics, 15, 12241-12300, 2015.

Regional Emission Inventory in Asia: http://www.jamstec.go.jp/frsgc/research/d4/emission.htm, 2007.

Geddes, J. A., and Martin, R. V.: Global deposition of total reactive nitrogen oxides from 1996 to 2014 constrained with satellite observations of NO2 columns, Atmos. Chem. Phys. Discuss., 2017, 1-44, 10.5194/acp-2016-1100, 2017.

Klimont, Z., Cofala, J., Schöpp, W., Amann, M., Streets, D. G., Ichikawa, Y., and Fujita, S.: Projections of SO2, NOx, NH3 and VOC Emissions in East Asia Up to 2030, Water, Air, & Soil Pollution, 130, 193-198, 2001.

Liu, L., Zhang, X., Wang, S., Lu, X., and Ouyang, X.: A Review of Spatial Variation of Inorganic Nitrogen (N) Wet Deposition in China, PloS one, 11, e0146051, 2016.

Liu, L., Zhang, X., Xu, W., Liu, X., Lu, X., Wang, S., Zhang, W., and Zhao, L.: Ground Ammonia Concentrations over China Derived from Satellite and Atmospheric Transport Modeling, Remote Sensing, 9, 467, 2017a.

Liu, L., Zhang, X., Zhang, Y., Xu, W., Liu, X., Zhang, X., Feng, J., Chen, X., Zhang, Y., Lu, X., Wang, S., Zhang, W., and Zhao, L.: Dry Particulate Nitrate Deposition in China, Environmental Science & Technology, 10.1021/acs.est.7b00898, 2017b.

Liu, X., Zhang, Y., Han, W., Tang, A., Shen, J., Cui, Z., Vitousek, P., Erisman, J. W., Goulding, K., and Christie, P.: Enhanced nitrogen deposition over China, Nature, 494, 459-462, 2013.

Nowlan, C., Martin, R., Philip, S., Lamsal, L., Krotkov, N., Marais, E., Wang, S., and Zhang, Q.: Global dry deposition of nitrogen dioxide and sulfur dioxide inferred from space‐based measurements, Global Biogeochemical Cycles, 28, 1025-1043, 2014.

Olivier, J. G. J., Bouwman, A. F., Hoek, K. W. V. D., and Berdowski, J. J. M.: Global air emission inventories for anthropogenic sources of NO x , NH 3 and N 2 O in 1990, Environmental Pollution, 102, 135-148, 1998.

Russell, A. R., Perring, A. E., Valin, L. C., and Bucsela, E. J.: A high spatial resolution retrieval of NO 2 column densities from OMI: method and evaluation, Atmospheric Chemistry & Physics, 11, 12411-12440, 2011.

Streets, D. G., Bond, T. C., Carmichael, G. R., Fernandes, S. D., He, D., Klimont, Z., Nelson, S. M., Tsai, N. Y., and Wang, M. Q.: An inventory of gaseous and primary aerosol emissions in Asia in the year 2000, Journal of Geophysical Research Atmospheres, 108, GTE 30-31, 2003.

Sun, Q., and Wang, M.: Ammonia Emission and Concentration in the Atmosphere over China, Scientia Atmospherica Sinica, 1997.

Wang, S. W., Liao, J. H., Yu-Ting, H. U., and Yan, X. Y.: A Preliminary Inventory of NH_3-N Emission

and Its Temporal and Spatial Distribution of China, Journal of Agro-Environment Science, 2009.

Wang, W. X., Lu, X. F., Pang, Y. B., Tang, D. G., and Zhang, W. H.: Geographical distribution of NH3

emission intensities in China, Actaentiae Circumstantiae, 1997.

Xia, Y., Zhao, Y., and Nielsen, C. P.: Benefits of China's efforts in gaseous pollutant control indicated

by the bottom-up emissions and satellite observations 2000–2014, Atmospheric Environment, 136,

43-53, http://dx.doi.org/10.1016/j.atmosenv.2016.04.013, 2016.

---

## Author Comment (AC2) · 16 Jun 2017

**Short comments: Dr. M. Van Damme**

We are grateful to Dr. M. Van Damme for his time and energy in providing helpful comments and guidance that have improved the manuscript. Detailed responses to the comments are given below (in blue).

This manuscript presents time-series of $NH_3$ total columns from the IASI satellite instrument. As the authors point out, the retrieved columns for this trace gas are strongly dependent on the meteorological information that is used as input data in the retrieval algorithm (in particular the surface temperature and the temperature profile in the lowest layers of the atmosphere). At present, the publically available IASI-$NH_3$ product uses the meteorological parameters as provided in EUMETSAT IASI Level 2 data.

The IASI Level 2 data is processed in real time, and there have been several updates to the algorithm since the launch of IASI. We can confirm that in particular, the change from version 5 to version 6 of the EUMETSAT Level 2 data on 30 September 2014 (and not November as stated in the manuscript), causes on average a substantial increase of the retrieved atmospheric NH3 columns. Hence the conclusions of the authors regarding NH3 trends should be taken with great care.

We thank M. Van Damme for pointing out the incorrect description in the manuscript, and have corrected it by changing "November 2014" to "September 2014".

---

## Author Comment (AC3) · 16 Jun 2017

**Referee #2**

We are grateful to the reviewer for the time and energy in providing helpful comments and guidance that have improved the manuscript. In this document, we describe how we have addressed the reviewer's comments. Detailed responses to each comment are given below (in blue).

This manuscript showed interesting results on the temporal evolution of $NO_x$ and $NH_3$ over China. By comparing the data resulting from inventories of REAS and EDGAR, the authors found that $NH_3$ and $NO_x$ continually increased over China during 1980-2010. Furthermore, based on previous satellite observations and an atmospheric chemistry transport model (MOZART-4), they also found that $NO_2$ over China increased from 2005 to 2011 and then decreased significantly from 2011 to 2015. Finally the authors discussed the plausible reasons including control policies of Chinese government to the emission trends of reactive nitrogen. Overall the topic of the study is sound and the manuscript was written well. However, I have the following concerns to be addressed before recommending it for publication in Atmos. Chem. Phys.

**Major comments:**

1. In line 168 of page 8, the authors filtered the DOMINO product with an absolute error below $10^{15}$ molecules $cm^{-2}$. However the $NO_2$ vertical column densities (VCDs) error depend on the net values of $NO_2$ VCDs. Therefore the filter may arbitrarily exclude the high $NO_2$ VCD values. The authors should evaluate the influence of absolute errors on the final emission results and show it in current study.

We used the DOMINO $NO_2$ product developed by Boersma et al. (2011). The fundamental algorithm of the retrieved $NO_2$ columns are the residual of subtracting two large numbers (the total slant column, and stratospheric slant $NO_2$ column). Because high $NO_2$ columns with high absolute errors as well as negative (or zero) $NO_2$ columns are statistically meaningful, they should not be discarded, as described

in the user manual (http://www.temis). In this revision, we did not filter the DOMINO product to perform the analysis.

2. The authors compared the emission data of $NO_2$ and $NH_3$ from satellite observations to that from Mozart-4 model simulations. But the authors did not explain whether the satellite overpass time has been considered during the comparison or not. The OMI satellite only gives the $NO_2$ data at about 1:30 pm of local time. The same time could also be used for the extraction of $NO_2$ data from Mozart-4 model. Whether this will influence the output results and conclusions of current study? This point should be clarified more.

Thank you very much for this good suggestion. In this revision, we have added the temporal trend analysis of $NO_2$ and $NH_3$ columns at 12:00 from MOZART to compare with that gained from satellite (OMI 1:45 P.M. local time) as shown in **Fig. 5, since the MOZART outputs vary over six hours** (00, 06, 12 and 18 h)**.**

We gained very similar results between OMI $NO_2$ (13:45 P.M.) and MOZART $NO_2$ at 12:00 with an increase rate of 4.02% $y^{-1}$ vs 4.23% $y^{-1}$ before 2011 and a decrease rate of -2.93% $y^{-1}$ (OMI) vs -3.07% $y^{-1}$ (MOZART) between 2011 and 2015 (Fig. 5). In general, we found an agreement on the $NO_2$ temporal trend between MOZART (12:00) and OMI (13:45) (refer to **Paragraph 3 in Sect. 3.3**).

[Figure]

**Fig. 5.** Time series of MOZART $NO_2$ and $NH_3$ columns over China during average warm months (April-September) and cold months (October-March) from 2008 to 2015. The mean columns were calculated by averaging the columns at 00, 6, 12 and 18 h. The associated mean error for each period is presented here as error bars.

3. The MOZART-4 model contained 12 bulk aerosol compounds, 39 photolysis, 85 gas species as well as 157 gas-phase reactions. However, the authors did not discuss the influence of $NO_x$ and $NH_3$ sink on their emission values at all while elucidating the data from MOZART-4. Although the authors have discussed the potential impacts of emission regulation or energy efficiency enhancement relevant government control policies on the $NO_x$ and $NH_3$ emissions, they are encouraged to show their insight on the correlations of atmospheric process of $NO_x$ and $NH_3$ with their final emission values.

Thank you very much for this good suggestion. $NH_3$ is the most abundant alkaline gas in the

troposphere and is important for its ability to neutralize acidic components such as sulfuric acid ($H_2SO_4$) and nitric acid ($HNO_3$) which form, respectively, by oxidation of emissions of sulfur dioxide ($SO_2$) and nitrogen oxides ($NO_x$). Reactions of $HNO_3$ and $H_2SO_4$ with $NH_3$ generally form submicron ammonium nitrate ($NH_4NO_3$) and ammoniated sulfate ($NH_4HSO_4$, $(NH_4)_2SO_4$, or other forms) particles. High temperatures also promote dissociation of $NH_4NO_3$ back to gaseous $NH_3$ and $HNO_3$. Therefore, the temporal trends of $NH_3$ and $NO_2$ should have an interactive impact between each other.

We have discussed the potential correlations of atmospheric process of $NO_x$ and $NH_3$ on the impact of the temporal trends in the following text added in **Paragraph 4 in Sect. 3.3** :

"In MOZART-4, the alkaline gaseous $NH_3$ and the acidic gaseous $NO_2$ (the precursor for $HNO_3$) and $SO_2$ are very important precursors for bulk $NH_4NO_3$ and $(NH_4)_2SO_4$ particles, which form the primary system of gas-particle partitioning ($NH_3$-$NH_4^+$-$NO_x$-$NO_3^-$-$SO_2$-$SO_4^{2-}$). The chemical shifts between particulate $NH_4NO_3$ and gaseous $NH_3$ and $NO_x$ are correlated with the abundance of $NH_3$ and $NO_x$ and meteorological factors. The decreased abundance of $NO_x$ between 2011 and 2015 may also contribute to an increase in the $NH_3$ abundance in the gas stage resulting from decreased conversion to particulate $NH_4NO_3$"

4. In section 3.1, the authors showed the emission data result from REAS and EDGAR, but they did not give convincing reasons for the different results of 0.24 kg N $ha^{-1}$ $y^{-2}$ from EDGAR and 0.17 kg N $ha^{-1}$ $y^{-2}$ from REAS. The authors should supply plausible explanations (e.g. induced by methodological difference of data compiling or meteorological factors etc.) to this. In addition, the authors thought 0.24 kg N $ha^{-1}$ $y^{-2}$ from EDGAR was much higher than 0.17 kg N $ha^{-1}$ $y^{-2}$ from REAS in lines 221-222 of page 11. However, they thought 0.33 kg N $ha^{-1}$ $y^{-2}$ was close to 0.24 kg N $ha^{-1}$ $y^{-2}$ in lines 231-232 of the same page. This is logically wrong. They need to correct it and also the relevant discussions.

We have added the possible reasons for the discrepancy between REAS and EDGAR as the following text in Sect. 3.1:

"The discrepancy in the magnitude of $NH_3$ increase rate from REAS and EDGAR (0.24 kg N ha$^{-1}$ y$^{-2}$ vs 0.17 kg N ha$^{-1}$ y$^{-2}$) in China since 1980 may come from the different emission factors considered for estimating $NH_3$ emissions. The EDGAR v4.3.1 $NH_3$ emissions were calculated based on sectors of agriculture, shipping, waste solid and wastewater, energy for buildings, process emissions during production and application, power industry, oil refineries, transformation industry, combustion for manufacturing, road transportation, railways, pipelines and off-road transport, while the REAS v1.1 $NH_3$ emissions focused mostly on the agriculture source (i.e., manure management of livestock and fertilizer application) (Crippa et al., 2015;Ohara et al., 2007). Moreover, the fundamental methodology of estimating the REAS v1.1 $NH_3$ emissions did not consider the seasonal agricultural variations compared with that of EDGAR v4.3.1 $NH_3$ emissions (Kurokawa et al., 2013), and the removal efficiency (as a key element used to estimate $NH_3$ emissions) in REAS v1.1 was also reported to be much higher than that in EDGAR v4.3.1 (Kurokawa et al., 2013).".

In addition, we have rewritten the sentences, which were logically wrong as the reviewer pointed out, by the following text at **Paragraph 3 in Sect. 3.1**:

"A previous study (Liu et al., 2013) summarized published data on the national anthropogenic $NH_3$ and $NO_x$ emissions with multi-periods in China (Wang et al., 2009;Wang et al., 1997;Streets et al., 2003;Klimont et al., 2001;Sun and Wang, 1997;Olivier et al., 1998;FRCGC, 2007), and also analyzed the temporal pattern of $NH_3$ emissions. Their results showed that the $NH_3$ emissions had increased at an annual average rate of 0.32 Tg N y$^{-2}$ (about 0.33 kg N ha$^{-1}$ y$^{-2}$). The increase rate of $NH_3$ emissions (0.33 kg N ha$^{-1}$ y$^{-2}$) by Liu et al. (2013) was double that in REAS (0.17 kg N ha$^{-1}$ y$^{-2}$), implying the

NH$_3$ increase rate in China is still an open question, and should be further studied ".

5. In lines 311-315 of page 15, the whole daily coverage over China cannot be achieved also due to the row anomaly effect. This effect may cause half of the satellite pixels to be unusable. The discussions here should be rearranged.

Thank you very much for this good suggestion. We have added the description of row anomaly effect and rearranged this discussion by the following text:

"For daily NO$_2$, the spatial coverage gained by OMI were influenced by cloud radiance fractions, surface albedo, solar zenith angles, row anomaly and so on (Russell et al., 2011;De Smedt et al., 2015). "Row anomaly" issue resulting from the OMI instrumental problem had an impact on approximately half of the rows undergoing unpredictable patterns in cross-track directions relying on latitudes and seasons and prevented obtaining convincing daily product with continuous coverage (Boersma et al., 2011;Boersma et al., 2016).".

6. Lines 99-101: the authors are encouraged to expand introduction on the method for converting satellite data to NH$_3$ column. Only a reference citation is not convenient for readers to follow up the work in a straight way.

We have expanded the introduction on the method of converting satellite data to NH$_3$ column by adding the following text:

"The retrieval algorithm of obtaining the IASI NH$_3$ total columns was based on the method in Whitburn et al. (2016). Two main steps were performed to derive the NH$_3$ columns from the satellite observations. First, deriving the spectral hyperspectral range index (HRI) based on each IASI observations (Walker et al., 2011;Van Damme et al., 2014). Second, converting HRI to NH$_3$ columns based on a constructed neural network with input parameters including vertical NH$_3$ profile, satellite

viewing angel, surface temperature and so on (Whitburn et al., 2016)".

**Minor comments:**

7. Line 102: the words of 'provides' and 'potential' should be changed to 'provide' and 'possibility'.

We have changed it as suggested.

8. Line 104: the description of 'emission data are also very important tools' is confusing, and there is no logic comparability with 'satellite observations' in the front dialogue, so I suggest to remove the 'tools' or modify the front dialogue properly.

We have removed the "tools".

9. Line 110: change 'resolutions' to 'resolution'.

We have changed it as suggested.

10. Line 170: change 'the manuscript' to 'previous work'.

We have changed it as suggested.

11. Line 130: change 'denotes' to 'denote'.

We have changed it as suggested.

12. Line 228-29: Similar information of the first dialogue here has been shown in lines 221-222, so there is no necessary to show it twice.

We have removed Line 228-229 to avoid repetition.

13. Line 229-230: the description of 'Liu et al. (2013) conducted that emissions of national anthropogenic NH3 and NOx summarized from published data during 1980-2010' is confusing and should be rearranged.

We have rewritten these sentences by the following text:

"A previous study (Liu et al., 2013) summarized published data on the national anthropogenic $NH_3$ and

$NO_x$ emissions with multi-periods in China (Wang et al., 2009;Wang et al., 1997;Streets et al., 2003;Klimont et al., 2001;Sun and Wang, 1997;Olivier et al., 1998;FRCGC, 2007), and also analyzed the temporal pattern of $NH_3$ emissions. Their results showed that the $NH_3$ emissions had increased at an annual average rate of 0.32 Tg N $y^{-2}$ (about 0.33 kg N $ha^{-1}$ $y^{-2}$). The increase rate of $NH_3$ emissions (0.33 kg N $ha^{-1}$ $y^{-2}$) by Liu et al. (2013) was double that in REAS (0.17 kg N $ha^{-1}$ $y^{-2}$), implying that the $NH_3$ increase rate in China is still an open question, and should be further studied in future work.".

14. Figure 1: add error bars to panel b please

Figure 1 shows a descriptive statistic of observation numbers by year, and we do not have error bars.

**Other corrections**

Removed original Fig. 6.

Since the information on the increase rate (%) between 2014 and 2015 from MOZART and IASI has been added in Fig. 3 and Fig. 5 in this revision, we have removed the original Fig. 6 to avoid duplication.

**References**

[revised manuscript text omitted]

---

## Author Response (AR2)

**Co-Editor Decision: Publish subject to technical corrections (10 Jul 2017) by Jennifer G. Murphy**

Comments to the Author:

I encourage the authors to consider the comments from Reviewer 1 regarding the revised version of the manuscript. In particular, it would be beneficial to have more clarity about which version of the $NH_3$ dataset was used and to what extent revisions might impact the trend analysis. Once this issue and the other details have been addressed, the manuscript is ready for publication.

Dear Dr. Murphy:

We have addressed the comments raised by Referee 1, and incorporated the comments/suggestions in the revised manuscript. In addition, we have also added detailed descriptions in Sect. 2.2 to clarity about which version of the $NH_3$ dataset was used and to what extent revisions might impact the trend analysis (lines 191-194).

Thank you very much for your consideration.

Sincerely,

Xiuying Zhang

On behalf of all co-authors

**Referee #1**

The authors have made an effort to address reviewer comments. Most technical issues have been addressed, but I have several suggestions for further revision.

We are grateful to the reviewer for the time and energy in providing helpful comments and guidance that have improved the manuscript. In this document, we describe how we have addressed the reviewer's comments. Detailed responses to each comment are given below (in blue).

1) The authors state in their response that this analysis is important because of the discussion on "the possible interactions between $NO_2$ and $NH_3$". However, this really isn't included or addressed in the manuscript. What exactly have the authors learned about this interaction, given the current results of their work?

Please refer to lines 276-286 in the main text. $NH_3$ is an important alkaline gas that is abundant in the atmosphere, and is able to neutralize acidic components including $HNO_3$ and $H_2SO_4$ through the oxidation of $NO_x$ and $SO_2$. We here just show a possible reason for an increase in $NH_3$ columns in recent years due to decreased acidic gases ($NO_2$ and $SO_2$) under strong control policy in 12-th FYP over China.

2) The authors mention the update in the NH3 retrieval in September 2014, but it's unclear whether they are actually using this updated retrieval. Are they? If so, please explicitly state this. If they aren't, they must include a comment (in response to the comment by Van Damme in the open discussion) on how they expect this update could affect their conclusions about NH3 trends, if at all.

We have added the following text in Sect. 2.2 for clarifications:

"We did not use the IASI $NH_3$ after September 30 in 2014 for the trend analysis because an update of the input meteorological data on 30 September 2014 has caused a substantial increase of the retrieved

atmospheric NH$_3$ columns."

Specific suggestions:

The trends are now included in %/yr. However, these are linear trends. So the authors must explain what this is relative to... Is it a % based on the first year of data? Or of the long term mean? First, NO2 increases by 4% per year to 2011, then decreases by 3.6 % per year to 2015... These % increases/decreases are very hard to compare when we don't know what the % is relative to. The authors must be more careful to explicitly explain how trends were calculated in their methods.

We have added the following text in Fig. 3 captions for clarifications:

"The percent increase or decrease rate (%) was the long term mean calculated by $[100 \times (\frac{Y_2 - Y_1}{Y_1} + \frac{Y_3 - Y_2}{Y_2} + \cdots + \frac{Y_{n+1} - Y_1}{Y_n})] \times \frac{1}{n}$."

1) Section 2.3, the authors write "...although their thread values of 0.24 kg N ha-1 yr-1...". What is a "thread" value? This wording is not clear to me.

We have added "(the slope in Fig. 2)" after "thread" for clarification.

2) I think the trend lines in the top panels of Figure 3a and 3b should be removed, since they are no longer commenting on the linear trend using monthly means.

We have removed them as suggested.

As a further suggestion, it's strange having two different ranges on the same plot for the same species (bottom panels) For example, NO2 ranges from 1.5-3 on the left side, but from 1-7 on the right side). I suggest two separate narrow panels so that we can actually distinguish the lines.

The different ranges on the same plot are mainly because the different scales of NO$_2$ ranges in warm (April-September) and cold (October-March) months.

To distinguish these two lines, we have changed the range of right Y-axis to "from 1 to 5".

[revised manuscript text omitted]